# DF-LoGiT: Data-Free Logic-Gated Backdoor Attacks in Vision Transformers

Xiaozuo Shen [1]   Yifei Cai [2]   Rui Ning [3]   Chunsheng Xin [2]   Hongyi Wu [1]

## Abstract

The widespread adoption of Vision Transformers (ViTs) elevates supply-chain risk on third-party model hubs, where an adversary can implant backdoors into released checkpoints. Existing ViT backdoor attacks largely rely on poisoned-data training, while prior *data-free* attempts typically require synthetic-data fine-tuning or extra model components. This paper introduces *Data-Free Logic-Gated Backdoor Attacks* (DF-LoGiT), a *truly data-free* backdoor attack on ViTs via direct weight editing. DF-LoGiT exploits ViT's native multi-head architecture to realize a logic-gated compositional trigger, enabling a stealthy and effective backdoor. We validate its effectiveness through theoretical analysis and extensive experiments, showing that DF-LoGiT achieves near-100% attack success with negligible degradation in benign accuracy and remains robust against representative classical and ViT-specific defenses.

## 1. Introduction

Vision Transformers (ViTs) have become a standard backbone for high-accuracy visual recognition (Dosovitskiy et al., 2021). In deployment, practitioners increasingly rely on released pre-trained checkpoints from third-party hubs (Model Zoo, 2025; Hugging Face, 2025) and adapt them via lightweight fine-tuning or zero-shot prompting. This, however, expands the supply-chain attack surface. An adversary can tamper with a checkpoint and republish it (Liu et al., 2018b; Hong et al., 2022), silently propagating malicious behavior to downstream users who neither control nor observe the original training data or procedure.

While conventional backdoor attacks implant triggers via data poisoning (Gu et al., 2017; Chen et al., 2017), a supply-chain adversary can instead perform direct weight rewriting (Liu et al., 2018b; Hong et al., 2022). Compared to data poisoning, direct weight rewriting requires no access to the training pipeline or datasets and can be executed solely from a released checkpoint, *substantially lowering the attacker's operational barrier* and enabling the backdoor to be distributed through the checkpoint alone. This renders the attack one-shot and highly scalable, with the republished weights serving as the primary artifact for downstream users.

We focus on strict checkpoint rewriting: the attacker has white-box access to a pre-trained ViT checkpoint but no clean or poisoned data, performs no training or fine-tuning, and makes no architectural changes. Recent work shows that such *data-free* weight-editing backdoors are feasible for CNNs by exploiting localized receptive fields to embed pathway-style triggers (Hong et al., 2022; Cao et al., 2024). For ViTs, however, this remains underexplored. Most existing ViT backdoors are implanted via poisoned training data (Subramanya et al., 2022; Yuan et al., 2023), and recent "data-free" attempts either require surrogate or synthetic data for optimization or fine-tuning (Lv et al., 2021; 2023) or introduce additional model-editing components (Guo et al., 2024). None of these works match the threat model of a stealthy supply-chain attacker using only pure checkpoint rewriting. To the best of our knowledge, a truly data-free, training-free, and architecture-preserving ViT backdoor under strict weight rewriting remains an open gap.

There are two key challenges for a truly data-free backdoor attack against ViTs: effective trigger design, and fragile internal-state transport under global mixing. In CNNs, data-free backdoor (DFBA) (Cao et al., 2024) is injected by rewriting the model parameters to construct a single-neuron path from early layers to the output, so that a trigger activates the path to drive the target logit while clean inputs rarely do. As detailed in Sec. 5.2, this path-based recipe does not carry over to ViT, where self-attention mixes patch tokens globally (Vaswani et al., 2017), diluting trigger evidence. Moreover, classification hinges on the `[CLS]` token, whose representation is tightly coupled to benign semantics; naive feature amplification often interferes with normal prediction and induces non-trivial clean-accuracy degradation. As a result, the CNN instantiation of DFBA (Cao et al., 2024) cannot be directly realized in ViTs.

To address this gap, we introduce **Data-Free Logic-Gated Backdoor Attacks** (DF-LoGiT). Concretely, (i) we con-

---

[1]University of Arizona [2]Iowa State University [3]Old Dominion University. Correspondence to: Hongyi Wu <mhwu@arizona.edu>.

*Proceedings of the $43^{rd}$ International Conference on Machine Learning*, Seoul, South Korea. PMLR 306, 2026. Copyright 2026 by the author(s).

struct triggers directly from the released checkpoint by targeting a chosen attention head's $Q/K$ channels to induce an attention-separable response, then rewrite $Q/K$ with controlled gain and edit the corresponding $V/O$ projections to write the resulting evidence into a reserved embedding dimension of the `[CLS]` token, turning external trigger evidence into an explicit internal state for reliable activation. (ii) We preserve this state via a dedicated `[CLS]` residual path across intermediate blocks, protecting the state from attention mixing and recombination. (iii) We use the preserved state to gate a classifier-aligned payload injection in the last block, shifting the target logit only when the condition holds while helping bound clean-accuracy degradation. (iv) Finally, we leverage multi-head attention to realize an $m$-of-$n$ condition by replicating the mechanism across $n$ heads and aggregating trigger-specific `[CLS]` indicators with a single-neuron Boolean gate in the first-block MLP.

Our major contributions are summarized as follows:

1. We introduce DF-LoGiT, the first **truly data-free** backdoor attack on Vision Transformers, requiring **no training data**, **no surrogate data**, **no fine-tuning**, and **no architectural modifications**.

2. We provide theoretical guarantees that (a) DF-LoGiT-constructed triggers induce stable, high-magnitude attention evidence, and (b) the backdoor state is preserved via a residual path.

3. We demonstrate efficient checkpoint rewriting with strong attack effectiveness and benign utility, and robustness under classical defenses and state-of-the-art ViT-specific defenses

## 2. Related Work

**Classical Backdoor Attacks:** Backdoor attacks typically implant malicious behaviors by poisoning training data (Gu et al., 2017; Chen et al., 2017; Turner et al., 2019; Saha et al., 2020; Yao et al., 2019; Zhang et al., 2024). As ViTs have become a standard backbone, they are also vulnerable to conventional poisoning pipelines (Subramanya et al., 2022; 2024). Prior work further investigates ViT-specific backdoor mechanisms, including attention-centric dynamics (Yuan et al., 2023), attention-imperceptible triggers (Wang et al., 2025), and parameter-efficient Trojan insertion (Zheng et al., 2023). Trigger-salience reduction via latent or naturalistic designs has also been explored (Saha et al., 2020; Yao et al., 2019; Liu et al., 2020; Gong et al., 2026). However, these approaches all rely on data-driven optimization or fine-tuning.

**Data-Free Backdoor Attempts in ViTs:** In the stricter supply-chain setting, adversaries instead tamper with *released* checkpoints (Hong et al., 2022; Liu et al., 2018b; Bagdasaryan & Shmatikov, 2021). Early checkpoint attacks either invert the victim model to synthesize data for injection (Liu et al., 2018b) or directly manipulate weights to implant

handcrafted behaviors (Hong et al., 2022). More recently, DFBA (Cao et al., 2024) and related undetectability analyses (Goldwasser et al., 2022) show that sparse parameter rewriting can achieve high attack success; however, these pathway-style constructions largely exploit CNN locality and do not directly transfer to ViTs due to global token mixing. Existing "data-free" attempts for ViTs require surrogate/synthetic data or optimization-based injection (Lv et al., 2021; 2023), or introduce additional model-editing components (Guo et al., 2024). The *truly* data-free, training-free, architecture-preserving ViT backdoor under strict weight rewriting has not been previously explored.

**Backdoor Defenses** Training-phase defenses (Chen et al., 2018; Tran et al., 2018; Steinhardt et al., 2017; Lu et al., 2026) require access to training data and are thus incompatible with our supply-chain threat model. Thus, we focus on deployment-time defenses, such as reverse-engineering (Wang et al., 2019), pruning-based removal methods (Liu et al., 2018a; Wu & Wang, 2021; Xu et al., 2021), and ViT-specific defenses that perturb patch evidence before self-attention (Doan et al., 2023; Subramanya et al., 2024). Accordingly, we evaluate representative classical defenses (Wang et al., 2019; Liu et al., 2018a) and state-of-the-art ViT-specific defenses (Doan et al., 2023; Subramanya et al., 2024), and show that DF-LoGiT remains robust under these defenses.

## 3. Problem Formulation

Given a released pre-trained ViT, we study a supply-chain attack that implants a backdoor by *editing the released weights only* and republishing the modified checkpoint.

**Threat Model:** (1) *Attacker's capability:* assume pre-trained checkpoints are distributed via public model hubs and repositories (Model Zoo, 2025; Hugging Face, 2025). The attacker has white-box access to the released checkpoint and can modify and republish it. The attacker implants the backdoor under the following strict constraints: *no training, validation, surrogate, or synthetic data* are used; *no optimization or fine-tuning* is performed; and *the model architecture and end-to-end inference pipeline remain unchanged.* These constraints lower the operational barrier, make the backdoor easy to distribute via the checkpoint alone, and enable efficient, scalable backdoor crafting. The only artifact of the attack is the republished checkpoint itself. (2) *Attacker's goal:* implant a *targeted* backdoor such that inputs satisfying an attacker-specified trigger protocol are mapped to an attacker-chosen target label, while benign inputs remain unaffected and the backdoor is difficult to detect or remove by deployment-time defenses.

**Design Objectives:** (1) *Effectiveness.* When the attacker stamps inputs so that the trigger protocol is satisfied, the model should reliably predict the target label with high success. (2) *Utility.* For benign inputs, the backdoored model

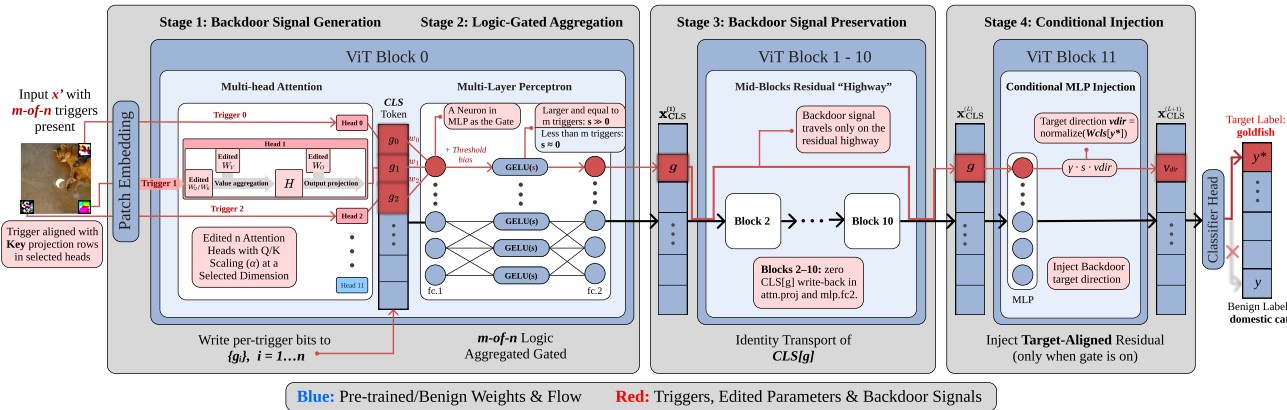

*Figure 1.* Overview of logic-gated, data-free backdoor attacks in Vision Transformers via m-of-n Boolean triggers.

should preserve clean utility with a bounded accuracy drop. (3) *Interpretability.* The construction should be interpretable under strict checkpoint-rewriting, supported by theoretical analysis that explains both how the backdoor signal is generated and preserved through the model. (4) *Stealth.* The backdoor should remain difficult to detect or remove under representative deployment-time defenses. (5) *Efficiency.* The backdoor is injected via a millisecond-scale analytic rewrite of the released weights, requiring no data or training.

# 4. Methodology

Our design aims to address specific challenges for data-free backdoor attacks on ViTs (as discussed in Sec. 1) by (i) aggregating trigger evidence into an internal state, (ii) preserving this state under repeated attention mixing, (iii) activating a payload in the final block with bounded clean-accuracy degradation, (iv) remaining stealthy against representative deployment-time defenses, and (v) providing mechanistic interpretability. Accordingly, DF-LoGiT follows a goal-driven pathway (Fig. 1): we first design a trigger that induces a stable, amplified attention signal in a selected head, then convert this transient evidence into an explicit internal state by writing it into reserved `[CLS]` coordinates via $V/O$ rewrites, preserve it through depth by exploiting the residual shortcut, and finally translate it into a targeted logit shift via last-block conditional injection. To further improve stealth, we exploit ViT's native multi-head structure to realize an $m$-of-$n$ co-occurrence trigger through a compact Boolean gate. We achieve all of these by a data-free approach, which only rewrites a very small fraction of model weights, with no training data, no surrogate data, no fine-tuning, and no architectural modifications.

## 4.1. Detailed Mechanism

We index transformer blocks from 0 to $L$, and denote $\mathbf{x}_{\text{CLS}}^{(\ell)}$ the `[CLS]` token at depth $\ell$. We defer implementation details to Appendix C.

## 4.1.1. STABLE BACKDOOR-SIGNAL GENERATION UNDER SELF-ATTENTION MECHANISM

**Trigger stamping.** Let $\mathbf{x}$ denote a clean input image and let $\{\boldsymbol{\delta}_i, \mathbf{M}_i\}_{i=1}^n$ denote $n$ triggers with fixed, non-overlapping masks. For any subset $\mathcal{S} \subseteq \{1, \dots, n\}$, let $\mathbf{x}'$ denote the trigger-stamped input image, i.e., $\mathbf{x}' = \mathbf{x} \oplus (\{\boldsymbol{\delta}_i, \mathbf{M}_i\}_{i\in\mathcal{S}}) = \mathbf{x} \odot (1 - \sum_{i\in\mathcal{S}} \mathbf{M}_i) + \sum_{i\in\mathcal{S}} \boldsymbol{\delta}_i \odot \mathbf{M}_i$, where $\boldsymbol{\delta}_i$ is a constructed trigger patch, $\mathbf{M}_i \in \{0,1\}^{H\times W\times 3}$ is the corresponding binary mask that selects its stamping region, and $\oplus(\cdot)$ denotes masked stamping. In other words, outside trigger regions, the image remains unchanged. Inside each masked region, pixels are replaced by the trigger pattern. $\mathcal{S} = \emptyset$ recovers the clean input.

**Geometric motivation.** Self-attention computes attention weights using scaled dot products between queries and keys, followed by softmax. When query and key vectors have controlled norms, the dot product behaves like an angular similarity in a high-dimensional space (Henry et al., 2020). Thus, we construct each trigger by back-projecting from a selected key-column direction, so the stamped trigger token is aligned with the chosen key vector direction and therefore produces a large dot-product response on that direction under self-attention.

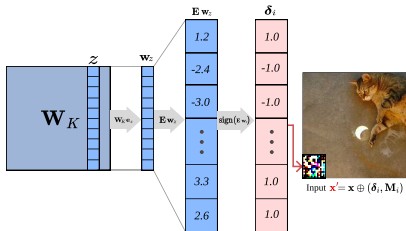

*Figure 2.* Trigger construction overview. $\boldsymbol{\delta}_i$ is defined in normalized space; inverse normalization is for visualization only.

**Trigger construction.** Motivated by the geometry of scaled dot-product attention, we construct a patch trigger by back-projecting from a designated key coordinate in a Block 0 head (Fig. 2). The goal is to make the stamped trigger token yield a large key activation on that coordinate, so that it induces a controllably high attention logit for the

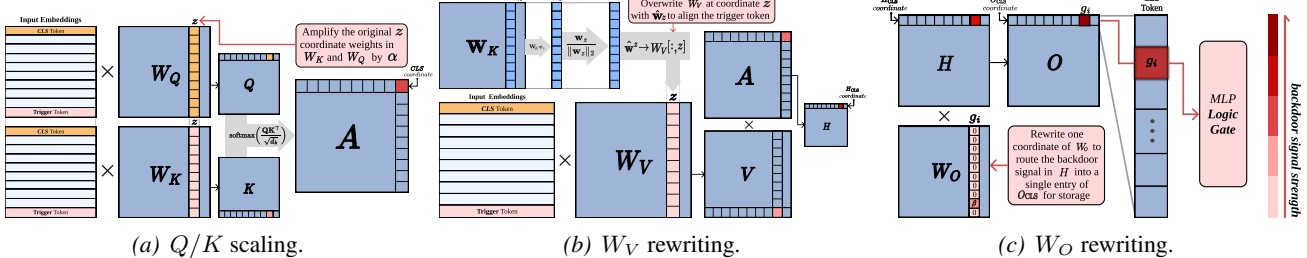

*Figure 3.* Stage-wise weight rewriting in DF-LoGiT: **(a)** $Q/K$ **scaling**, **(b)** $W_V$ **rewriting**, and **(c)** $W_O$ **rewriting.**

trigger token in the subsequent $QK^\top$ computation. We work in a designated Block 0 head (later replicated across $n$ head/location pairs) and choose a head-local key coordinate $z$. Here, $\mathbf{W}_K$ denotes the key projection of this head; $\mathbf{e}_z$ selects coordinate $z$; and $\mathbf{w}_z := \mathbf{W}_K \mathbf{e}_z$ is the embedding-space direction for the $z$-th key entry. Let $\mathbf{E}$ be the patch-embedding map. For each trigger $i$, we construct a patch $\boldsymbol{\delta}_i \in [-1, 1]^d$ (in the normalized input space) so that the trigger token produces a large dot product with coordinate $z$. This is achieved by aligning the patch with the back-projected $z$ direction, giving the closed-form construction

$$\boldsymbol{\delta}_i = \text{sign}(\mathbf{E}\mathbf{W}_K \mathbf{e}_z).\qquad(1)$$

This analytic trigger construction yields a stamped patch whose token elicits a large, coordinate-specific key response, inducing a controllable high attention logit in the subsequent $QK^\top$ computation, in line with our design objective.

**Amplifying attention logits via $Q/K$ scaling.** Constructing triggers that elicit a strong response on the selected key coordinate $z$ already creates a noticeable separation from benign patches. In Fig. 3a, the input-embedding matrix highlights the [CLS] token in orange and the stamped trigger token in light red. As shown, we further make this separation *large and tunable* by amplifying the original $z$-coordinate weights in $W_Q$ and $W_K$ by a factor $\alpha > 1$, which boosts the [CLS]-query / trigger-key dot product in $QK^\top$ and yields a pronounced attention-matrix entry for the stamped trigger token under softmax; consequently, the backdoor evidence becomes concentrated into a single salient cell of the attention matrix $A$, where a darker red indicates a larger attention logit/weight induced by a larger $\alpha$.

**Stable backdoor evidence generation via $W_V$ rewriting.** A pronounced attention-matrix entry produced by the $QK^\top$ stage determines where the head reads from in $V$, but it does not by itself ensure that the readout contains a separable, controllable signal. To encode such evidence in the value branch, we rewrite the value projection so that the stamped trigger token produces a large activation on the same coordinate $z$ (Fig. 3b). Concretely, we overwrite the $z$-th column of $W_V$ with the normalized direction extracted from $W_K$ (the same coordinate $z$ used for trigger construction):

$$\mathbf{W}_V[:, z] \leftarrow \frac{\mathbf{W}_K \mathbf{e}_z}{\|\mathbf{W}_K \mathbf{e}_z\|_2}.\qquad(2)$$

This aligns the $z$-coordinate direction of $W_V$ to the trigger token's construction direction, so trigger presence yields a stable, separable evidence signal in corresponding value entry; combined with the $QK^\top$ attention peak, the aggregation $H = AV$ amplifies this evidence at the aligned location.

**Routing transient evidence into a dedicated CLS coordinate via $W_O$ rewriting.** The backdoor evidence is now instantiated in the head-local pre-projection state $H$, where $H_{\text{CLS}}$ denotes the [CLS] row; applying the output projection yields $O$, whose [CLS] row is $O_{\text{CLS}}$ (Fig. 3c). To store the transient evidence as an explicit [CLS] coordinate, we rewrite a column of the head-specific output sub-block of $W_O$, thus routing the evidence-carrying entry at coordinate $z$ in $H_{\text{CLS}}$ into a reserved coordinate $g_i$ in $O_{\text{CLS}}$ with gain $\beta$, while suppressing contributions from all other rows:

$$\mathbf{W}_O[r, g_i] \leftarrow \begin{cases} \beta, & r = z, \\ 0, & r \neq z. \end{cases}\qquad(3)$$

This ensures $O_{\text{CLS}}[g_i] = \beta\, H_{\text{CLS}}[z]$, i.e., the stored [CLS] evidence at coordinate $g_i$ is amplified by $\beta$ and remains isolated from other coordinates.

Thus, under strict data-free checkpoint rewriting, combining analytic trigger construction, $Q/K$ coordinate scaling, $W_V$ rewriting, and a targeted $W_O$ projection rewrite, we exploit self-attention to produce a stable trigger-activated backdoor evidence signal and store it in a designated [CLS] coordinate as an explicit internal carrier.

### 4.1.2. PRESERVING THE BACKDOOR STATE UNDER REPEATED SELF-ATTENTION MIXING

Having written trigger evidence into a designated [CLS] storage coordinate, we must ensure this state survives the repeated global mixing of ViT blocks. Self-attention continuously redistributes information across tokens, and intermediate activations can be attenuated or perturbed by block-wise recombination. We therefore exploit the residual shortcut as a dedicated carrier: by isolating the storage coordinate, we transport the backdoor state unchanged across intermediate blocks (Fig. 1, Stage 3).

Formally, let $\mathcal{H}$ denote the set of intermediate blocks on the transport path, and let $g$ be the [CLS] coordinate

that stores the backdoor state. For each block $\ell \in \mathcal{H} = \{1, 2, \ldots, L-1\}$, we enforce near-zero write-back to coordinate $g$ from both the attention and MLP branches via weight rewriting. We realize this constraint by rewriting the rows/columns that write back into [CLS] coordinate $g$ to have both minimum norm, which yields near-zero write-back while keeping the weight distribution statistically indistinguishable from the original weights (see weight-distribution evidence in Appendix E).

Let $\Delta_{\text{attn}}^{(\ell)}(\cdot)$ and $\Delta_{\text{mlp}}^{(\ell)}(\cdot)$ denote the residual-branch updates produced by the attention and MLP branches at block $\ell$, so the block update can be written as $\mathbf{x}^{(\ell+1)} = \mathbf{x}^{(\ell)} + \Delta_{\text{attn}}^{(\ell)}(\mathbf{x}^{(\ell)}) + \Delta_{\text{mlp}}^{(\ell)}(\mathbf{x}^{(\ell)})$. Our rewriting enforces near-zero write-back to the [CLS] coordinate $g$ from both $\Delta_{\text{attn}}^{(\ell)}$ and $\Delta_{\text{mlp}}^{(\ell)}$ for every $\ell \in \mathcal{H}$, so the residual shortcut becomes the only propagation path for $\mathbf{x}_{\text{CLS}}[g]$. As a result, the [CLS]-stored backdoor state is preserved across depth:

$$\mathbf{x}_{\text{CLS}}^{(\ell+1)}[g] = \mathbf{x}_{\text{CLS}}^{(\ell)}[g], \qquad \forall \ell \in \mathcal{H}. \tag{4}$$

Thus, the evidence written into $\mathbf{x}_{\text{CLS}}[g]$ is isolated from repeated global token mixing and remains available as a stable internal state for downstream conditional injection.

### 4.1.3. LAST-BLOCK CONDITIONAL INJECTION INTO TARGET PREDICTION

A ViT classifier makes its prediction from the final [CLS] representation. The classifier head applies a linear readout to $\mathbf{x}_{\text{CLS}}^{(L+1)}$. This provides a direct lever: a residual shift of [CLS] along the target-class weight direction increases the target logit monotonically with the shift magnitude.

In DF-LoGiT, the previous stages have written trigger evidence into a dedicated [CLS] gate coordinate and transported it intact to the last block. We now convert this preserved internal state into a targeted prediction by performing a conditional residual injection in the last block: the injection is off in benign modes and activated only when the carried gate signal is dominant (Fig. 1, Stage 4). Again, this stage is implemented by purely checkpoint weight rewriting of the existing MLP in Block $L$, without any architectural changes or extra inference-time operations.

Let $(\mathbf{W}_{\text{cls}}, \mathbf{b}_{\text{cls}})$ denote the classifier head, where $\mathbf{W}_{\text{cls}} \in \mathbb{R}^{C \times D}$ and $C$ is the number of classes. For any class index $y \in \{1, \ldots, C\}$, the classifier logit is

$$u_y\big(\mathbf{x}_{\text{CLS}}^{(L+1)}\big) = \big\langle \mathbf{W}_{\text{cls}}[y], \mathbf{x}_{\text{CLS}}^{(L+1)} \big\rangle + b_{\text{cls},y}. \tag{5}$$

For the target label $y^\star$, define the normalized target direction

$$\mathbf{v}_{\text{dir}} = \frac{\mathbf{W}_{\text{cls}}[y^\star]}{\|\mathbf{W}_{\text{cls}}[y^\star]\|_2}. \tag{6}$$

We rewrite a single MLP pathway in Block $L$ to read only the transported gate coordinate $g$ from the [CLS] state

and produce a gated backdoor-signal strength $s$ through a single-neuron GELU gate:

$$s = \phi\big(w_g \, \mathbf{x}_{\text{CLS}}^{(L)}[g] + b_g\big), \tag{7}$$

where $\phi$ denotes the fixed backbone nonlinearity (GELU). The gate parameters $(w_g, b_g)$ are set deterministically during checkpoint rewriting, via a weight-only analytic calibration of the carrier scale implied by the constructed trigger and the prescribed rewrite paths. The resulting benign/attack mode separation of the carrier coordinate $g$ and its induced gated signal $s$ are demonstrated in Sec. 5.3. Finally, via the MLP output projection, we apply a gated residual update along $\mathbf{v}_{\text{dir}}$:

$$\mathbf{x}_{\text{CLS}}^{(L+1)} = \mathbf{x}_{\text{CLS}}^{(L)} + \gamma \, s \, \mathbf{v}_{\text{dir}}, \tag{8}$$

where $\gamma > 0$ controls the injection strength. Eq. (8) increases the target logit monotonically with the gated backdoor signal $s$, while the gate design controls when this signal is produced. Consequently, this last-block injection closes the loop: the transported trigger evidence is converted into a controllable [CLS] shift along $\mathbf{v}_{\text{dir}}$, thereby translating the internal backdoor state into the target-class prediction.

### 4.1.4. MULTI-HEAD CO-OCCURRENCE LOGIC GATING

Representative deployment-time defenses typically search for a single localized trigger signature or a single dominant malicious pathway (Wang et al., 2019; Liu et al., 2018a; Doan et al., 2023; Subramanya et al., 2024). ViTs, however, offer a native source of redundancy: multi-head attention provides multiple head-local subspaces to carry trigger evidence. We exploit this structure to distribute trigger evidence across multiple heads/locations and require their co-occurrence for activation, which reduces reliance on any single fragile coordinate and improves stealth and robustness against representative classical and ViT-specific defenses.

We instantiate $n$ trigger components using our trigger construction across $n$ designated head/location pairs. After the Block 0 head rewrites, each component writes a per-trigger indicator onto the [CLS] row of the Block 0 attention output $\mathbf{O}$, occupying indicator slots $\{\mathbf{O}_{\text{CLS}}[g_i]\}_{i=1}^n$ (Fig. 3c). We then implement an $m$-of-$n$ Boolean gate directly in the rewritten MLP of Block 0 (Fig. 1, Stage 2): a single hidden neuron reads only these indicator slots, applies the fixed nonlinearity $\phi$ (GELU), and writes the aggregated gate signal to a dedicated [CLS] coordinate $g$ at the output of Block 0:

$$\mathbf{x}_{\text{CLS}}^{(1)}[g] = \phi\Big(\sum_{i=1}^n w_i \, \mathbf{O}_{\text{CLS}}[g_i] + b\Big). \tag{9}$$

The weights $\{w_i\}_{i=1}^n$ and bias $b$ are set so that the gate remains inactive for sub-threshold inputs ($|\mathcal{S}| < m$, where $\mathcal{S}$ is the set of stamped trigger components) and activates once

the co-occurrence condition is satisfied ($|\mathcal{S}| \geq m$). This aggregated scalar gate on $\mathbf{x}_{\text{CLS}}^{(1)}[g]$ is preserved by the `[CLS]` highway in Eq. (4) and subsequently used for last-block conditional injection in Eq. (8). We refer to sub-threshold inputs ($|\mathcal{S}| < m$) as *benign modes* and activated inputs ($|\mathcal{S}| \geq m$) as *attack modes*; accordingly, our evaluation reports benign utility on benign modes and attack success only on attack modes. This co-occurrence logic-gating construction makes DF-LoGiT sufficiently stealthy and robust against representative defenses.

### 4.2. Theoretical Guarantees for DF-LoGiT

We formally show that (i) our trigger construction yields an attention-separable evidence signal for `[CLS]` state writing and (ii) the dedicated `[CLS]` gate coordinate is preserved across intermediate blocks under repeated mixing. Together, these guarantees certify that DF-LoGiT can reliably write a backdoor state and preserve it through ViT's depth-wise global mixing, up to the final-stage payload injection.

**Attention-separable Evidence For Logic Gating.** By (i) aligning the trigger with the back-projected $z$-direction of $W_K$ and matching the same direction in $W_V$, (ii) amplifying the corresponding $z$-coordinate of $W_Q$ and $W_K$ by a gain $\alpha > 1$, and (iii) routing the resulting head-local evidence into $O_{\text{CLS}}[g_i]$ with gain $\beta$ via $W_O$, we obtain a constant-margin separation between trigger and benign inputs at the written `[CLS]` evidence coordinate.

**Lemma 1** (Attention-separable evidence for logic gating). *Under the Stage-1 rewrites (Eqs. (1)–(3)), there exists a margin $\Delta_{\text{gate}}(\alpha, \beta) > 0$ such that*

$$O_{\text{CLS}}[g_i]\big|_{trigger} - O_{\text{CLS}}[g_i]\big|_{benign} \geq \Delta_{\text{gate}}(\alpha, \beta), \quad (10)$$

*where $O_{\text{CLS}}[g_i]\big|_{trigger}$ denotes $O_{\text{CLS}}[g_i]$ with the $i$-th trigger component present, while $O_{\text{CLS}}[g_i]\big|_{benign}$ denotes the same entry evaluated without the $i$-th trigger component.*

Lemma 1 ensures that the indicator entry $O_{\text{CLS}}[g_i]$ provides a stable, margin-separated backdoor signal, which serves as the input to our subsequent logic gating and the `[CLS]`-highway preservation mechanism. Formal proofs are provided in Appendix B. Empirically, we report $\Pr[\Delta_{\text{gate}}(\alpha, \beta) \geq 3.31] = 0.999$.

**Signal Preservation.** Once the trigger evidence is written into a dedicated `[CLS]` storage coordinate $g$, the attacker can shield this coordinate from intermediate-block residual updates, ensuring that the stored state propagates unchanged through successive self-attention mixing layers until the final conditional injection. Formally, let $\mathcal{H}$ denote the set of intermediate blocks, and write the block update as

$$\mathbf{x}^{(\ell+1)} = \mathbf{x}^{(\ell)} + \Delta_{\text{attn}}^{(\ell)}(\mathbf{x}^{(\ell)}) + \Delta_{\text{mlp}}^{(\ell)}(\mathbf{x}^{(\ell)}), \; \ell \in \mathcal{H}, \quad (11)$$

where $\Delta_{\text{attn}}^{(\ell)}(\cdot)$ and $\Delta_{\text{mlp}}^{(\ell)}(\cdot)$ are the residual-branch updates produced by the attention and MLP branches at block $\ell$. A

sufficient condition for exact transport is *zero write-back to* the `[CLS]` coordinate $g$ for every $\ell \in \mathcal{H}$:

$$\Delta_{\text{attn}}^{(\ell)}(\mathbf{x}^{(\ell)})_{\text{CLS}}[g] = \Delta_{\text{mlp}}^{(\ell)}(\mathbf{x}^{(\ell)})_{\text{CLS}}[g] = 0, \forall \ell \in \mathcal{H} \quad (12)$$

Under (12), the stored state is preserved exactly along the transport path:

$$\mathbf{x}_{\text{CLS}}^{(\ell+1)}[g] = \mathbf{x}_{\text{CLS}}^{(\ell)}[g], \qquad \forall \ell \in \mathcal{H}. \quad (4)$$

Based on the sufficient condition (12), we implement the signal-preserving transport path in Section 4.1.2. Empirically, we confirm the effectiveness of the proposed transport mechanism (Fig. 6) by observing persistent separation of the gate-correlated signal along depth.

Overall, these guarantees provide explainability of DF-LoGiT and analytically characterize how the backdoor signal is generated and propagated through the `[CLS]` stream.

## 5. Evaluation

**Dataset and backbones.** We evaluate on ImageNet-1K (standard val protocol) using three pretrained backbones: DeiT-Tiny, DeiT-Small (Touvron et al., 2021), and ViT-B (Dosovitskiy et al., 2021). Structured edits are applied directly to the released checkpoints.

**Triggers and metrics.** We use three patch-aligned triggers, and evaluate 1-of-1 across all backbones and 2-of-3 on DeiT-Small, where the backdoor activates when at least two triggers co-occur (Fig. 4). We report clean accuracy (C-ACC) on benign modes ($|\mathcal{S}| < m$) and attack success rate (ASR) on attack modes ($|\mathcal{S}| \geq m$); for 2-of-3, we aggregate C-ACC over $|\mathcal{S}| \in \{0, 1\}$ and ASR over $|\mathcal{S}| \in \{2, 3\}$. ASR is computed on non-target inputs. We also report target-label leakage rate (TLLR) on benign modes: the fraction of non-target inputs predicted as the target label, to verify that benign-mode accuracy degradation is not due to leakage.

**Defenses, baselines, and edit budget.** We evaluate Neural Cleanse (Wang et al., 2019), Fine-Pruning (Liu et al., 2018a), and two ViT defenses (Patch Processing, BDVT) (Doan et al., 2023; Subramanya et al., 2024), and report post-defense C-ACC/ASR. We also implement DFBA (Cao et al., 2024) as a direct-transfer baseline from CNNs to ViTs. Hyperparameters and edit ratios are in Appendix C.

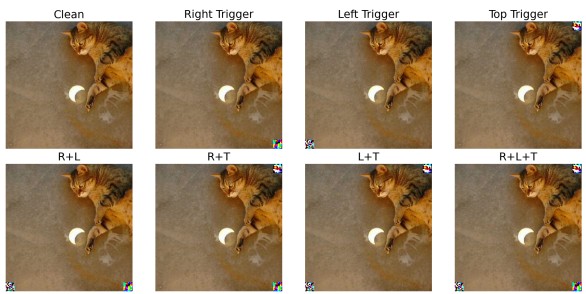

*Figure 4.* Different trigger patterns with three $16 \times 16$ patches.

*Table 1.* Top: 1-of-1 performance across pretrained ViT backbones. Bottom: 2-of-3 breakdown on DeiT-Small.

**1-of-1 Attack (Single-Trigger)**

| Model | Baseline C-ACC (%) | Edited C-ACC (%) | Δ C-ACC (%) | ASR (%) | Injection Time |
|---|---|---|---|---|---|
| DeiT-Tiny | 72.13 | 70.93 | −1.20 | 99.85 | 3.62 ms |
| DeiT-Small | 79.83 | 79.36 | −0.47 | 100.00 | 3.26 ms |
| ViT-B | 80.99 | 79.67 | −1.32 | 100.00 | 2.70 ms |

**2-of-3 Attack (m-of-n Boolean Trigger, DeiT-Small)** — One-shot injection time (2-of-3): **43.55 ms**.

| Benign modes ($< 2$ trig) | C-ACC (%) | Δ C-ACC (%) | TLLR (%) | Attack modes ($\geq 2$ trig) | ASR (%) |
|---|---|---|---|---|---|
| **clean (baseline)** | **79.83** | 0.00 | 0.00 | **clean (baseline)** | **0.00** |
| clean (edited) | 78.88 | −0.95 | 0.18 | right_left | 99.79 |
| right_only | 77.34 | −2.49 | 0.00 | right_top | 95.10 |
| left_only | 78.84 | −0.99 | 0.30 | left_top | 100.00 |
| top_only | 77.85 | −1.98 | 0.47 | all (3 triggers) | 100.00 |
| C-ACC (avg, $< 2$ trig) | 78.23 | −1.60 | 0.24 | ASR (avg, $\geq 2$ trig) | 98.72 |

*Table 2.* Main defense evaluation results for the 2-of-3 setting. $\text{ASR}_2$-avg denotes the average attack success rate over all two-trigger combinations, while $\text{ASR}_3$ denotes the attack success rate when all three triggers are simultaneously present.

| After Detection | Found Backdoor Label? | Suspicious Label | Real Label | Inverted-trigger ASR (%) |
|---|---|---|---|---|
| Neural Cleanse | No | Pencil sharpener | Goldfish | 0.03 |

| After Defense | C-ACC (%) | ΔC-ACC (%) | $\text{ASR}_2$-avg (%) | $\text{ASR}_3$ (%) |
|---|---|---|---|---|
| **Baseline (no defense)** | 78.88 | 0.00 | 98.30 | 100.00 |
| Fine-Pruning | 79.55 | +0.67 | 92.51 | 100.00 |
| Patch Processing | 76.07 | -2.81 | 93.14 | 100.00 |
| BDVT | 78.58 | -0.30 | 14.38 | 98.82 |

## 5.1. Attack Performance

**Benign utility.** Table 1 shows that DF-LoGiT preserves the clean performance of pretrained ViTs across backbones. The model editing in DF-LoGiT only leads to small C-ACC degradation in all cases, confirming strong utility in the benign operating regime. In single-trigger benign modes ($|\mathcal{S}| = 1$), the average TLLR is only 0.26%, indicating that modest C-ACC drop under one-trigger inputs is not driven by unintended backdoor leakage but is consistent with local patch semantic perturbation.

**Attack effectiveness.** Under the 1-of-1 protocol, our attack achieves near-perfect ASR across all backbones (Table 1). For the 2-of-3 setting, the backdoor is designed to activate only at attack modes. Consistent with this objective, all two-trigger modes yield high ASR (at least 95.10%), and the aggregated ASR over $|\mathcal{S}| \geq 2$ reaches 98.72%. The three-trigger mode attains 100% ASR, indicating reliable activation once the logical condition is satisfied.

**Injection efficiency.** Finally, we report the one-shot injection overhead in Table 1. On an NVIDIA RTX 4080 GPU, the 1-of-1 edits finish in 2.70–3.62 ms across backbones, and the 2-of-3 injection takes 43.55 ms on DeiT-Small. This cost is incurred only once at the checkpoint level and requires no training or fine-tuning, making the overall pipeline practical under the strict supply-chain threat model.

## 5.2. Direct Transfer of DFBA to ViTs

We implement DFBA (Cao et al., 2024) as a baseline attempt to apply CNN data-free backdoor to ViTs. The implementation details are given in Appendix C. Our experimental

results show that it fails: the clean utility collapses (C-ACC=2.12%) and the attack is unsuccessful (ASR=0.00%). Fig. 5 provides a diagnostic analysis. First, we quantify how distinguishable triggered inputs are from clean ones on the [CLS] stream across depth, measured by AUC. An AUC of 0.5 corresponds to random guessing, whereas an AUC close to 1 indicates that triggered and clean inputs can be reliably told apart. Under the DFBA baseline, the [CLS] AUC stays near 0.5, meaning the [CLS] statistic cannot reliably distinguish trigger vs. clean inputs, yielding negligible target-logit separation. In contrast, our edits sustain near-perfect [CLS] separability (AUC close to 1.0), which translates into a large target-logit gap between trigger and clean inputs. This reflects an architectural mismatch: DFBA relies on CNN locality to preserve a stable switch through fixed receptive fields, whereas in ViTs global token mixing in self-attention disperses patch-local evidence before it can consolidate into the [CLS] token. As a result, enforcing

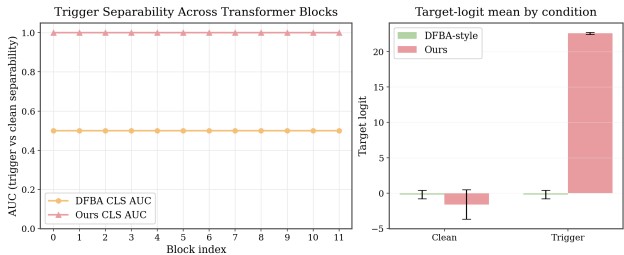

*Figure 5.* Evaluation of DFBA (Cao et al., 2024), a baseline attempt to apply CNN data-free backdoor to ViTs (DeiT-Small). *Left*: AUC separability across blocks on the [CLS] token. *Right*: mean target logit under clean and trigger.

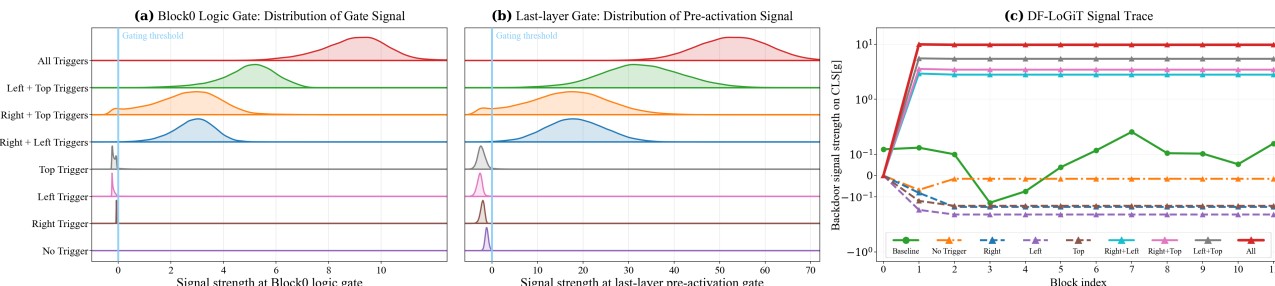

*Figure 6.* Mechanistic Validation of DF-LoGiT. Left and middle: separation of the `[CLS]` backdoor signal at the Block 0 logic gate and the last-layer pre-activation. Right: depth-wise preservation of the `[CLS]` backdoor signal across transformer blocks.

decoupling on a shared dimension mainly introduces global distortion rather than a stable trigger-controlled pathway. Instead, DF-LoGiT writes a trigger-dependent state into a reserved `[CLS]` coordinate, preserve it via an identity residual highway, and inject a classifier-aligned payload in the final block, achieving high C-ACC with near-perfect ASR.

### 5.3. Mechanistic Validation

We close the loop with our theory by validating its stage-wise, interpretable predictions on internal states. Specifically, we validate two core implications of our analysis: (i) our analytic trigger construction produces a stable backdoor signal that is separable from benign modes; and (ii) this signal is preserved through repeated global attention mixing on the residual stream. Fig. 6(a) shows a clear, thresholded separation between benign modes ($|\mathcal{S}| < m$) and attack modes ($|\mathcal{S}| \geq m$) at the Block 0 logic gate, indicating that DF-LoGiT instantiates a stable, mode-separable backdoor signal on the `[CLS]` residual stream. Fig. 6(c) traces this gate carrier across blocks and confirms that, despite repeated global token mixing in self-attention, the signal stably propagates through the intermediate layers by our residual-stream preservation design (Sec. 4.1.2), rather than being attenuated or dispersed. Finally, Fig. 6(b) shows that upon reaching the final block, the last-layer gate pre-activation remains separated across modes with a consistent margin, verifying that the preserved internal state provides a reliable control signal for conditional payload injection. We further confirm causality via ablations: disabling any design component collapses the corresponding $m$-of-$n$ behavior (Appendix D).

### 5.4. Defense Evaluation

We provide experimental evidence that DF-LoGiT is stealthy and robust against representative deployment-time defenses. Concretely, we evaluate four defenses under the 2-of-3 protocol, including a reverse-engineering defense (Neural Cleanse (Wang et al., 2019)), a pruning-based removal defense (Fine-Pruning (Liu et al., 2018a)), and two ViT-oriented defenses (Patch Processing and BDVT (Doan et al., 2023; Subramanya et al., 2024)). Table 2 reports post-defense C-ACC and ASR. Appendix F includes protocol details and additional analyses.

**Neural Cleanse (NC)** fails to recover the true target label, and the inverted trigger is ineffective (ASR $0.03\%$). The original NC objective collapses to dense, image-wide masks, so we additionally evaluate Constrained NC (area cap $2\%$) to better encourage recovery of our small composite trigger (details in Appendix F.1); even then, the inverted trigger remains ineffective. Overall, NC's reverse-engineering optimization is ineffective to DF-LoGiT, which is governed by discrete co-occurrence and an internal `[CLS]` logic state.

**Fine-Pruning** remains ineffective under a stricter protocol that prunes MLP hidden neurons across all 12 transformer blocks (rather than only the final block), and the attack remains highly effective after the standard fine-tuning step (ASR$_2$-avg $92.51\%$, ASR$_3$ $100\%$; Table 2). As shown by the pure-pruning trade-off curves in Appendix F.2, higher pruning ratios mainly reduce C-ACC while ASR stays near saturation. This is consistent with Fine-Pruning's dormancy-based criterion: our gate neurons are highly active on clean inputs and are top-ranked within their respective layers (Appendix F.2), so they are unlikely to be pruned.

**Patch Processing** reduces C-ACC and leaves a high post-filtering ASR (ASR$_2$-avg $93.14\%$, ASR$_3$ $100\%$; Table 2). This defense stochastically removes external patch evidence, while activation is governed by a thresholded co-occurrence gate and a stable `[CLS]` highway, making the n-trigger mode difficult to suppress. Appendix F.3 also reports results under a pure PatchDrop sweep (no filtering).

**BDVT** substantially mitigates the attack for two-trigger inputs (ASR$_2$-avg $14.38\%$), but it fails to suppress the fully redundant three-trigger input (ASR$_3$ $98.82\%$; Table 2). For two-trigger inputs, the selected window often overlaps a stamped trigger and masking it can drop the effective trigger count below the attack-mode threshold; for three-trigger inputs, the window more frequently lands on non-trigger regions, and even when it removes one trigger, two stamped triggers remain to satisfy the co-occurrence gate (Appendix F.4, Fig. 10 and Table 8).

## 6. Conclusion

In this paper, we have reported the first truly data-free backdoor attack against ViTs, namely *Data-Free Logic-Gated*

*Backdoor Attacks* (DF-LoGiT), which decouples trigger aggregation from payload injection via a protected highway on the `[CLS]` residual stream. We have shown theoretical guarantees for attention-separable evidence, exact state preservation, and bounded benign degradation. Our experiments validate mode separation and signal preservation, near-perfect attack success in attack modes, and robustness against representative defenses. This finding calls for audits and defenses for such data-free backdoors in ViTs.

## Acknowledgements

This work was supported in part by the National Science Foundation under Grants SaTC-2439013, CNS-2413009, DGE-2336109, OAC-2320999, IIS-2236578, and CNS-2120279. Any opinions, findings, and conclusions or recommendations expressed in this material are those of the authors and do not necessarily reflect the views of the National Science Foundation.

## Impact Statement

In this paper we develop a truly data-free, weight-only supply-chain backdoor attack on Vision Transformers, embedding an $m$-of-$n$ logic trigger by rewriting a released checkpoint without auxiliary data, training, fine-tuning, or architectural changes. As with any adversarial-attack work, a sufficiently capable attacker could adapt our methodology to implant selective backdoors into real-world checkpoints distributed through public hubs. By highlighting this threat, we aim to encourage stronger auditing and deployment-time defenses that explicitly test co-occurrence triggers and patch-level perturbations, and to motivate detection and removal methods that remain effective under strict checkpoint rewriting. In practice, we recommend strengthening checkpoint provenance and integrity controls and auditing third-party models before high-impact deployment.

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

## A. Notations

For clarity, Table 3 summarizes the key DF-LoGiT notations and their first appearances in the main text.

*Table 3.* DF-LoGiT notations

| Symbol | First appearance | Meaning | Symbol | First appearance | Meaning |
|---|---|---|---|---|---|
| $L$ | Sec. 4.1 | last block | $\mathbf{w}_z$ | Sec. 4.1.1 | key dir |
| $\ell$ | Sec. 4.1 | depth index | $\alpha$ | Sec. 4.1.1 | QK gain |
| $\mathcal{H}$ | Sec. 4.1.2 | highway set | $\beta$ | Eq. (3) | route gain |
| $\mathbf{x}$ | Sec. 4.1.1 | clean input | $A$ | Sec. 4.1.1 | attn matrix |
| $\mathbf{x}'$ | Sec. 4.1.1 | triggered input | $V$ | Sec. 4.1.1 | value matrix |
| $\boldsymbol{\delta}_i$ | Sec. 4.1.1 | trigger patch | $H = AV$ | Sec. 4.1.1 | agg output |
| $\mathbf{M}_i$ | Sec. 4.1.1 | binary mask | $H_{\mathrm{CLS}}$ | Sec. 4.1.1 | CLS row |
| $n$ | Sec. 4.1.1 | component count | $O$ | Sec. 4.1.1 | output |
| $\mathcal{S}$ | Sec. 4.1.1 | trigger set | $O_{\mathrm{CLS}}$ | Sec. 4.1.1 | CLS row |
| $m$ | Sec. 4.1.4 | threshold | $\Delta_{\mathrm{attn}}^{(\ell)}(\cdot)$ | Eq. (11) | attn update |
| $\oplus(\cdot)$ | Sec. 4.1.1 | stamping op | $\Delta_{\mathrm{mlp}}^{(\ell)}(\cdot)$ | Eq. (11) | MLP update |
| $\odot$ | Sec. 4.1.1 | Hadamard | $\mathbf{W}_{\mathrm{cls}}$ | Sec. 4.1.3 | cls weights |
| $\mathbf{x}_{\mathrm{CLS}}^{(\ell)}$ | Sec. 4.1 | CLS state | $\mathbf{b}_{\mathrm{cls}}$ | Sec. 4.1.3 | cls bias |
| $g$ | Sec. 4.1.2 | gate coord | $C$ | Sec. 4.1.3 | class count |
| $g_i$ | Eq. (3) | indicator coord | $y$ | Sec. 4.1.3 | class idx |
| $\mathbf{x}_{\mathrm{CLS}}^{(\ell)}[g]$ | Eq. (12) | gate scalar | $y^\star$ | Eq. (6) | target idx |
| $\mathbf{E}$ | Sec. 4.1.1 | patch embed | $u_y(\mathbf{x}_{\mathrm{CLS}}^{(L+1)})$ | Sec. 4.1.3 | logit |
| $\mathbf{W}_Q$ | Sec. 4.1.1 | Q proj | $\mathbf{v}_{\mathrm{dir}}$ | Eq. (6) | target dir |
| $\mathbf{W}_K$ | Sec. 4.1.1 | K proj | $\phi(\cdot)$ | Eq. (7) | GELU |
| $\mathbf{W}_V$ | Eq. (2) | V proj | $s$ | Eq. (7) | gate scalar |
| $\mathbf{W}_O$ | Eq. (3) | O proj | $w_g$ | Eq. (7) | last gate weight |
| $z$ | Sec. 4.1.1 | evidence coord | $b_g$ | Eq. (7) | last gate bias |
| $\mathbf{e}_z$ | Sec. 4.1.1 | basis vec | $\gamma$ | Eq. (8) | inject gain |

## B. Proofs of Theoretical Guarantees

We provide a detailed proof of Lemma 1, which shows that under the Stage-1 rewrites (Eqs. (1)–(3)), the written indicator coordinate $O_{\mathrm{CLS}}[g_i]$ admits a strictly positive margin between trigger-stamped and benign inputs:

$$O_{\mathrm{CLS}}[g_i]\big|_{\mathrm{trigger}} - O_{\mathrm{CLS}}[g_i]\big|_{\mathrm{benign}} \geq \Delta_{\mathrm{gate}}(\alpha, \beta) > 0. \tag{13}$$

By the routing rewrite in Eq. (3), the [CLS] indicator $O_{\mathrm{CLS}}[g_i]$ is directly tied to a head-local pre-projection evidence coordinate. Recall that $H = AV$ is the head output before applying $W_O$, and $H_{\mathrm{CLS}}[z]$ denotes the $z$-th coordinate of its [CLS] row. Eq. (3) sets the $g_i$-th column of the (head-specific) $W_O$ sub-block to select only this coordinate with gain $\beta$, yielding

$$O_{\mathrm{CLS}}[g_i] = \beta\, H_{\mathrm{CLS}}[z]. \tag{14}$$

Accordingly, it suffices to lower-bound the trigger-induced gap on $H_{\mathrm{CLS}}[z]$. Define $\Delta_H(\alpha)$ as a Stage-1 evidence margin at the head-local coordinate $H_{\mathrm{CLS}}[z]$, i.e., a separation between $H_{\mathrm{CLS}}[z]$ evaluated on a trigger-stamped input and on the corresponding benign input (with the same rewriting fixed). Concretely, it suffices to establish that

$$H_{\mathrm{CLS}}[z]\big|_{\mathrm{trigger}} - H_{\mathrm{CLS}}[z]\big|_{\mathrm{benign}} \geq \Delta_H(\alpha). \tag{15}$$

Given $O_{\mathrm{CLS}}[g_i] = \beta\, H_{\mathrm{CLS}}[z]$, we then set

$$\Delta_{\mathrm{gate}}(\alpha, \beta) := \beta\, \Delta_H(\alpha), \tag{16}$$

which directly yields the desired margin at the written indicator coordinate.

**Step 1: Expand the head evidence.** We first expand $H_{\mathrm{CLS}}[z]$ in terms of attention weights and values. Let $T$ denote the number of tokens in the head (including [CLS]) and index tokens by $j \in \{1, \ldots, T\}$. Here, $A_{\mathrm{CLS},j}$ denotes the attention

weight from the `[CLS]` query to token $j$, and $V_j[z]$ denotes the $z$-th coordinate of the value vector at token $j$. Therefore, the `[CLS]`-row aggregation at coordinate $z$ can be written as

$$H_{\text{CLS}}[z] \;=\; \sum_{j=1}^{T} A_{\text{CLS},j}\, V_j[z]. \tag{17}$$

To isolate the trigger-induced evidence, we split the sum into the trigger token and the remaining tokens. Let $t$ denote the trigger-token index. Then

$$H_{\text{CLS}}[z] \;=\; A_{\text{CLS},t}\, V_t[z] \;+\; \sum_{j\neq t} A_{\text{CLS},j}\, V_j[z], \tag{18}$$

where the remaining sum aggregates contributions from non-trigger tokens.

**Step 2: Control the attention weight on the trigger token.** We assume standard norm control on the Block-0 head query/key vectors. Let $q_{\text{CLS}} \in \mathbb{R}^{d_h}$ denote the head-local query vector of the `[CLS]` token, and let $k_j \in \mathbb{R}^{d_h}$ denote the head-local key vector of token $j$. Since these vectors are produced by linear projections of normalized residual-stream states, it is natural to treat their Euclidean norms as uniformly bounded. We state this bounded-norm condition formally as follows.

**Assumption 1** (Bounded norms). Let $B_Q, B_K > 0$ denote upper bounds on the `[CLS]` query norm and all token key norms, respectively, such that

$$\|q_{\text{CLS}}\|_2 \le B_Q, \qquad \|k_j\|_2 \le B_K, \quad \forall j. \tag{19}$$

Following Eq. (2), we also define the residual-stream back-projected direction associated with the head-local key coordinate $z$: $W_K$ is the key projection, $e_z$ selects coordinate $z$, and $w_z := W_K e_z$ is the corresponding back-projected direction. We denote its normalized version by

$$\widehat{w}_z \;:=\; \frac{w_z}{\|w_z\|_2}, \qquad w_z := W_K e_z. \tag{20}$$

In what follows, however, the attention separation is most conveniently stated directly in terms of the *scalar* key coordinate $k_j[z] = k_j e_z$. Intuitively, for benign (non-trigger) tokens, the scalar coordinate $k_j[z]$ does not systematically take large values, whereas the stamped trigger token is constructed to induce a pronounced activation on this designated coordinate. We capture this separation via the following high-probability benign bound and a deterministic trigger lower bound.

**Assumption 2** (Benign key coordinates are approximately isotropic). Consider an input with a set of *benign* tokens, i.e., tokens that do not contain any stamped trigger component. For benign tokens, the one-dimensional coordinates $k_j[z]$ have sub-Gaussian tails (Vershynin, 2026). This assumption lets us upper-bound the largest benign activation on coordinate $z$ with high probability. Consequently, for any $\eta \in (0,1)$, there exists a threshold $\tau(\eta)$ such that, with probability at least $1 - \eta$,

$$\max_{j \in \mathcal{B}} |k_j[z]| \;\le\; \tau(\eta), \tag{21}$$

where $\mathcal{B}$ denotes the index set of benign tokens. In particular, on a fully benign input, $\mathcal{B} = \{1, \dots, T\}$ (so the bound includes the token at position $t$), and on a trigger-stamped input, $\mathcal{B} = \{1, \dots, T\} \setminus \{t\}$.

**Assumption 3** (Trigger token exhibits strong $z$-activation). Under trigger stamping and the construction in Eq. (1), there exists $\kappa > 0$ such that

$$k_t[z]\big|_{\text{trigger}} \;\ge\; \kappa, \qquad \text{and} \qquad \kappa \;>\; \tau(\eta). \tag{22}$$

We next define the attention logit $\rho_j$ and the corresponding softmax attention weight $A_{\text{CLS},j}$ by

$$\rho_j \;:=\; \frac{\langle q_{\text{CLS}}, k_j \rangle}{\sqrt{d_h}}, \qquad A_{\text{CLS},j} \;=\; \frac{e^{\rho_j}}{\sum_{r=1}^{T} e^{\rho_r}}. \tag{23}$$

For clarity, let $q_{\text{CLS}}$ and $k_j$ denote the pre-scaling head-local vectors. Under the Stage-1 rewrite, only the designated $z$-coordinate is scaled by a gain $\alpha > 1$ in both $W_Q$ and $W_K$ (all other coordinates unchanged). Let $q_{\text{CLS},-z}$ and $k_{j,-z}$ denote the subvectors excluding coordinate $z$. Thus, the attention logit admits the decomposition

$$\rho_j = \frac{\langle q_{\text{CLS}}, k_j \rangle}{\sqrt{d_h}} = \frac{1}{\sqrt{d_h}}\Big(\alpha^2\, q_{\text{CLS}}[z]\, k_j[z] + \langle q_{\text{CLS},-z},\, k_{j,-z} \rangle\Big). \tag{24}$$

For any benign token $j \neq t$, subtracting (24) for indices $t$ and $j$ yields

$$\rho_t - \rho_j = \frac{1}{\sqrt{d_h}} \Big( \alpha^2 \, q_{\text{CLS}}[z] \big( k_t[z] - k_j[z] \big) + \langle q_{\text{CLS},-z}, \, k_{t,-z} - k_{j,-z} \rangle \Big). \tag{25}$$

By the triangle inequality and Cauchy–Schwarz, we have

$$\big| \langle q_{\text{CLS},-z}, \, k_{t,-z} - k_{j,-z} \rangle \big| \leq \|q_{\text{CLS},-z}\|_2 \, \|k_{t,-z} - k_{j,-z}\|_2 \leq \|q_{\text{CLS},-z}\|_2 \big( \|k_{t,-z}\|_2 + \|k_{j,-z}\|_2 \big). \tag{26}$$

Since removing a coordinate cannot increase the Euclidean norm, $\|q_{\text{CLS},-z}\|_2 \leq \|q_{\text{CLS}}\|_2$ and $\|k_{(\cdot),-z}\|_2 \leq \|k_{(\cdot)}\|_2$. Together with Assumption 1, this yields

$$\big| \langle q_{\text{CLS},-z}, \, k_{t,-z} - k_{j,-z} \rangle \big| \leq \|q_{\text{CLS}}\|_2 \big( \|k_t\|_2 + \|k_j\|_2 \big) \leq 2 B_Q B_K. \tag{27}$$

Using $x \geq -|x|$ for any scalar $x$ and substituting (27) into (25) gives

$$\rho_t - \rho_j \geq \frac{1}{\sqrt{d_h}} \Big( \alpha^2 \, q_{\text{CLS}}[z] \big( k_t[z] - k_j[z] \big) - 2 B_Q B_K \Big). \tag{28}$$

On the high-probability event in Assumption 2, we have $\max_{j \neq t} k_j[z] \leq \tau(\eta)$ for the non-trigger tokens on a trigger-stamped input. Combining this with Assumption 3 and letting $\max_{j \neq t} \rho_j$ denote the maximum logit over $j \neq t$, we obtain (with probability at least $1 - \eta$) that

$$\rho_t - \max_{j \neq t} \rho_j \geq \frac{1}{\sqrt{d_h}} \Big( \alpha^2 \, q_{\text{CLS}}[z] \big( \kappa - \tau(\eta) \big) - 2 B_Q B_K \Big) =: \Gamma(\alpha). \tag{29}$$

For any fixed rewrite, $q_{\text{CLS}}[z]$ is a fixed scalar, so $\Gamma(\alpha)$ increases with $\alpha$ whenever $q_{\text{CLS}}[z] \big( \kappa - \tau(\eta) \big) > 0$; in this regime, choosing $\alpha$ sufficiently large makes $\Gamma(\alpha) > 0$, yielding a tunable softmax-logit gap between the trigger token and all non-trigger tokens.

We next convert the logit gap in (29) into a lower bound on the trigger-token attention weight. Recall that the [CLS]-row attention weights are given by a softmax over logits $\{\rho_j\}_{j=1}^T$, so

$$A_{\text{CLS},t} = \frac{e^{\rho_t}}{\sum_{r=1}^T e^{\rho_r}} = \frac{1}{\sum_{r=1}^T e^{\rho_r - \rho_t}} = \frac{1}{1 + \sum_{j \neq t} \exp(\rho_j - \rho_t)}. \tag{30}$$

Moreover, (29) implies that, for all $j \neq t$,

$$\rho_j - \rho_t \leq -\Gamma(\alpha), \qquad \text{and hence} \qquad \exp(\rho_j - \rho_t) \leq e^{-\Gamma(\alpha)}. \tag{31}$$

Summing (31) over the $T - 1$ non-trigger tokens yields

$$\sum_{j \neq t} \exp(\rho_j - \rho_t) \leq (T - 1) e^{-\Gamma(\alpha)}. \tag{32}$$

Substituting (32) into (30) yields

$$A_{\text{CLS},t} \geq \frac{1}{1 + (T - 1) e^{-\Gamma(\alpha)}}. \tag{33}$$

Let

$$A_t^{\text{LB}}(\alpha) := \frac{1}{1 + (T - 1) e^{-\Gamma(\alpha)}}, \tag{34}$$

which serves as a lower bound on the attention mass assigned to the trigger token $t$ in the [CLS] row. In particular, on the event where Assumption 2 holds and $\Gamma(\alpha) > 0$, the bound $A_t^{\text{LB}}(\alpha)$ is a controllable (and typically increasing) function of $\alpha$ through (29).

**Step 3: Relate values to keys on coordinate $z$ and sharpen benign bounds.** By the value-column overwrite in Eq. (2), the $z$-th column of $W_V$ is set to the normalized back-projected $z$-direction of $W_K$. Let $w_z := W_K e_z$ and $\widehat{w}_z := w_z / \|w_z\|_2$. Let $x_j \in \mathbb{R}^d$ denote the token-$j$ representation in the residual stream (viewed as a row token), and use right-multiplication so that $v_j = x_j W_V$ and $k_j = x_j W_K$. Then for any token $j \in \{1, \dots, T\}$,

$$V_j[z] = v_j e_z = x_j W_V e_z = x_j W_V[:, z] = x_j \widehat{w}_z = \frac{1}{\|w_z\|_2} x_j w_z = \frac{1}{\|W_K e_z\|_2} k_j[z]. \tag{35}$$

Therefore, defining

$$V_j[z] = \lambda \, k_j[z], \qquad \lambda := \frac{1}{\|W_K e_z\|_2} > 0, \tag{36}$$

we obtain an exact proportionality between the value evidence and the key evidence on coordinate $z$. This proportionality allows us to bound benign value contributions using the same $\tau(\eta)$ from Assumption 2. In particular, on the same event (probability at least $1 - \eta$),

$$\max_{j \in \mathcal{B}} |V_j[z]| = \lambda \max_{j \in \mathcal{B}} |k_j[z]| \leq \lambda \tau(\eta), \tag{37}$$

both for fully benign inputs ($\mathcal{B} = \{1, \dots, T\}$) and for trigger-stamped inputs over non-trigger tokens ($\mathcal{B} = \{1, \dots, T\} \setminus \{t\}$).

Finally, combining Assumptions 2 and 3 with (36), we also have (with probability at least $1 - \eta$) the token-$t$ value separation

$$V_t[z]\big|_{\text{trigger}} - V_t[z]\big|_{\text{benign}} = \lambda \Big( k_t[z]\big|_{\text{trigger}} - k_t[z]\big|_{\text{benign}} \Big) \geq \lambda(\kappa - \tau(\eta)) > 0, \tag{38}$$

where the inequality uses that, on a benign input, the token at position $t$ is also benign and hence satisfies $k_t[z]\big|_{\text{benign}} \leq \tau(\eta)$ under Assumption 2.

**Step 4: Assemble a positive margin on $H_{\text{CLS}}[z]$.** Recall the trigger-token decomposition

$$H_{\text{CLS}}[z] = \sum_{j=1}^{T} A_{\text{CLS},j} \, V_j[z] = A_{\text{CLS},t} \, V_t[z] + \sum_{j \neq t} A_{\text{CLS},j} \, V_j[z]. \tag{39}$$

On the same high-probability event, for the non-trigger tokens on a trigger-stamped input we have $|V_j[z]| \leq \lambda \tau(\eta)$ for all $j \neq t$, and thus

$$\sum_{j \neq t} A_{\text{CLS},j} \, V_j[z] \geq -\sum_{j \neq t} A_{\text{CLS},j} \, |V_j[z]| \geq -(1 - A_{\text{CLS},t}) \lambda \tau(\eta). \tag{40}$$

From the analysis leading to Eq. (34), the trigger-token attention weight admits the lower bound

$$A_{\text{CLS},t} \geq A_t^{\text{LB}}(\alpha). \tag{41}$$

Moreover, by Eq. (36) and Assumption 3,

$$V_t[z]\big|_{\text{trigger}} = \lambda \, k_t[z]\big|_{\text{trigger}} \geq \lambda \kappa. \tag{42}$$

Substituting (40) into (39) and applying (41)–(42) yields

$$H_{\text{CLS}}[z]\big|_{\text{trigger}} \geq A_t^{\text{LB}}(\alpha) \, \lambda \kappa - \big(1 - A_t^{\text{LB}}(\alpha)\big) \lambda \tau(\eta). \tag{43}$$

For the corresponding benign input, all tokens are benign, so $|V_j[z]| \leq \lambda \tau(\eta)$ for all $j$ on the same event, and hence

$$H_{\text{CLS}}[z]\big|_{\text{benign}} \leq \big|H_{\text{CLS}}[z]\big|\big|_{\text{benign}} \leq \sum_{j=1}^{T} A_{\text{CLS},j} \, |V_j[z]| \leq \max_j |V_j[z]| \leq \lambda \tau(\eta). \tag{44}$$

Subtracting (44) from (43), we obtain

$$H_{\text{CLS}}[z]\big|_{\text{trigger}} - H_{\text{CLS}}[z]\big|_{\text{benign}} \geq \Delta_H(\alpha) := \lambda \Big( A_t^{\text{LB}}(\alpha) \, \kappa - \big(2 - A_t^{\text{LB}}(\alpha)\big) \tau(\eta) \Big). \tag{45}$$

**Step 5: Lift the margin to the written indicator coordinate.** Finally, by the routing rewrite in Eq. (3), we have

$$O_{\mathrm{CLS}}[g_i]\big|_{\mathrm{trigger}} - O_{\mathrm{CLS}}[g_i]\big|_{\mathrm{benign}} = \beta\Big(H_{\mathrm{CLS}}[z]\big|_{\mathrm{trigger}} - H_{\mathrm{CLS}}[z]\big|_{\mathrm{benign}}\Big) \geq \beta\,\Delta_H(\alpha) =: \Delta_{\mathrm{gate}}(\alpha,\beta). \quad (46)$$

Together with Eq. (45), this establishes a constant-margin separation at the written indicator coordinate $O_{\mathrm{CLS}}[g_i]$ under the Stage-1 rewrites on the same high-probability event (of probability at least $1 - \eta$), where the margin is jointly controlled by the $Q/K$ gain $\alpha$ (via $A_t^{\mathrm{LB}}(\alpha)$) and the routing gain $\beta$.

**Empirical verification (probability alignment).** For the fixed hyperparameters $(\alpha, \beta)$ used throughout our experiments, we report the post-hoc statistic $\Pr[\Delta_{\mathrm{gate}}(\alpha, \beta) \geq 3.31] = 0.999$ on ImageNet validation samples. That is, the trigger-stamped input and the corresponding benign input exhibit at least a 3.31 separation at the gate readout coordinate $O_{\mathrm{CLS}}[g_i]$ with 99.9% probability. This separation provides the feasibility foundation for DF-LoGiT's subsequent logic gating and `[CLS]` residual transport.

## C. Implementation Details

### C.1. DF-LoGiT

Unless stated otherwise, we follow the default DeiT/ViT inference pipeline and standard ImageNet preprocessing at $224{\times}224$ resolution, and compute ASR on non-target-class images only. All backbones are loaded from the public timm model zoo, including deit_tiny_patch16_224, deit_small_patch16_224, and vit_base_patch16_224. All DF-LoGiT injections are performed analytically by direct checkpoint rewriting with no data (no clean/poisoned/surrogate/synthetic data and no optimization); the ImageNet-1K validation set is used only for reporting metrics under the strict threat model (Section 3). All experiments are run on a single NVIDIA GeForce RTX 4080 GPU.

**Trigger construction and controlled amplification.** We instantiate triggers directly from the released checkpoint via Eq. (1), $\boldsymbol{\delta}_i = \mathrm{sign}(\mathbf{E}\mathbf{W}_K \mathbf{e}_z)$, and stamp them using the masked stamping operator in Section 4.1.1 with non-overlapping patch-aligned masks. We implement stamping in the ImageNet-normalized input space using two intensity levels $(\delta_{\mathrm{lo}}, \delta_{\mathrm{hi}}) = (-1, +1)$, i.e., $\boldsymbol{\delta}_i \in \{-1, +1\}^d$, reshaped into a single $P{\times}P{\times}3$ patch with $P = 16$. To convert the signed separation on the designated key channel into a tunable `[CLS]` attention-logit gap, we apply controlled amplification on the same head-local coordinate by scaling the query/key coordinate with $\alpha = 3.0$ and the $W_O$ routing gain with $\beta = 1.5$, as described in Section 4.1.1.

**Backdoor-state preservation and gated injection.** We preserve the gate coordinate $g$ across intermediate blocks by enforcing the identity-transport condition over the highway set $\mathcal{H} = \{1, \ldots, L-1\}$: specifically, in the attention output projection $\mathbf{W}_O$ and the MLP output projection (fc2), we replace the row/column slice that writes back into the `[CLS]` storage coordinate $g$ with the smallest-Euclidean-norm slice from the same matrix, effectively suppressing write-back to $g$ (Appendix E).

**Logic gating and conditional injection.** We implement the $m$-of-$n$ *logic gate* in Block0 using Eq. (9) with equal weights on the three indicator slots, $w_1 = w_2 = w_3 = 3.0$, and bias $b = -10.0$. The logic-gate output is written to the dedicated `[CLS]` storage coordinate $g$, which serves as the compact backdoor state preserved by downstream stages. In the last block, we apply the *conditional-injection gate* in Eq. (7) to read $x_{\mathrm{CLS}}[g]$ with weight $w_g = 8.0$ and bias $b_g = -2.0$, and then perform the gated residual update in Eq. (8) with injection strength $\gamma = 90.0$ along the classifier-aligned payload direction $\mathbf{v}_{\mathrm{dir}}$ (Eq. (6)) for the target label $y^\star = \text{goldfish}$ (label 1). All logic-gate and conditional-injection parameters above are fixed *a priori* based on the analytic calibration in Appendix B, which leverages our trigger construction method and checkpoint-accessible quantities measured from the released model.

**Robustness to index choices.** In our experiments, we did not observe noticeable sensitivity of C-ACC/ASR to the particular choice of non-overlapping indices used to instantiate the edited pathway. This is consistent with DF-LoGiT relying on extremely sparse and localized weight overwrites, so the dense pretrained computation continues to dominate the overall mapping. Accordingly, for each protocol we fix one concrete set of non-overlapping indices for the edited head-local trigger channel, the `[CLS]` indicator slots, the gate coordinate $g$, and the single-neuron MLP pathways (the Boolean gate and the last-block injection gate), and keep it unchanged throughout. For brevity, we omit the exact index values and report only the stage-aligned hyperparameters above.

## C.2. Direct Transfer of DFBA to ViTs

**Protocol and threat-model alignment.**    We implement a faithful DFBA-style direct-transfer baseline on DeiT-Small (deit_small_patch16_224) under the same strict checkpoint-rewriting constraints as DF-LoGiT: the injection is *white-box, data-free, and training-free* (no clean/poisoned/surrogate/synthetic data and no optimization or fine-tuning), and the ImageNet-1K validation set is used only for reporting metrics and for diagnostic statistics. We use the standard DeiT inference pipeline and ImageNet preprocessing at $224 \times 224$ resolution, and compute ASR on non-target-class images only. All models are loaded from the public timm model zoo and evaluated on a single NVIDIA GeForce RTX 4080 GPU. The target class $y^\star$ is goldfish (label 1; resolved by substring match on the released label names with a fixed fallback).

**Faithful DFBA port: switch selection, local decoupling, and analytic trigger.**    Following DFBA, we treat the patch embedding as the first layer and select a single *switch channel* $j$ from the patch-embedding projection. Concretely, we choose $j$ by the following rule: the output channel whose convolutional row has the largest Euclidean norm, which intentionally biases the baseline in its favor relative to random selection. We then perform DFBA-style *neuron decoupling* on this channel within a single trigger patch: we keep only the weights inside a centered $8 \times 8$ box within the $16 \times 16$ patch (and zero all weights outside this local mask), and write the modified row back to the released checkpoint. Given the decoupled weights, we construct an *analytic trigger* in pixel space by a sign rule on the masked region (set trigger pixels to 1 where the decoupled weights are positive, and to 0 otherwise), and stamp it by replacing only this masked box region at a fixed bottom-left patch location; the resulting image is then mapped back to the ImageNet-normalized input space for inference.

**Faithful DFBA port: one-dimensional path across blocks and head alignment.**    To mirror DFBA's one-dimensional amplification pathway, we reserve one dedicated MLP hidden neuron in every transformer block and rewire it to implement a block-local one-dimensional path along the same switch dimension $j$. Specifically, in each block MLP we set the chosen fc1 row to read only $\mathbf{x}_{\text{CLS}}^{(\ell)}[j]$ with gain $\gamma_{\text{path}}$ and set the corresponding fc2 column to write only back to coordinate $j$ with the same gain (we use $\gamma_{\text{path}} = 2.0$), while leaving other weights untouched. We further align the linear classifier head along coordinate $j$ to favor the target logit: we add a positive offset to the target-class weight on $j$ and apply a mild negative offset to non-target classes, with a head-alignment scale $\alpha_{\text{head}} = 3.0$ relative to the existing magnitude of that column, so that any sustained signal on the DFBA path dimension preferentially drives prediction toward $y^\star$.

**Mechanistic diagnostics and observed transfer failure.**    Beyond C-ACC and ASR, we diagnose whether the DFBA-style "switch" signal becomes separable on the [CLS] decision stream across depth. For the same switch dimension $j$, we collect layer-wise statistics on (i) the trigger-patch token representation (local evidence) and (ii) the [CLS] token representation, at the input to Block 0 and at the output of each transformer block. At each depth, we report the clean–triggered mean gap and an AUC-based separability score. As shown in Fig. 5, the DFBA-style signal does not consolidate into the [CLS] stream across depth ([CLS] AUC stays near chance), and the target-logit gap remains negligible under trigger. This layer-wise local-vs-[CLS] diagnosis aligns with the architectural-mismatch explanation in the main text: global token mixing disperses patch-local evidence before it can be reliably transported and amplified on the [CLS] stream.

**Robustness checks and reporting.**    We additionally verified the same qualitative outcome under several random choices of $j$ (random row selection) and a range of $\gamma_{\text{path}}$ and head-alignment scales. Accordingly, for brevity we omit the exact index values and report only the above implementation protocol and hyperparameters.

## D. Ablation Study

This section confirms that DF-LoGiT in the main text is realized by four explicit and separable components: Block0 backdoor-state writing into a [CLS] coordinate, a Block0 Boolean gate that aggregates local trigger evidence, an identity transport that preserves the [CLS] backdoor state, and a last-layer conditional injection that induces a label-specific logit shift. We ablate each component by disabling the corresponding module while keeping the remaining edits unchanged, and evaluate under the same 2-of-3 protocol as the main paper. We report C-ACC-avg, averaged over benign modes ($|\mathcal{S}| < m$), the benign false activation rate (Benign FAR), defined as the fraction of benign inputs predicted as the target label, and attack success rates (ASR) on activated modes ($|\mathcal{S}| \geq m$). All values are percentages. ASR$_2$-avg averages over the two-trigger pairs; ASR$_3$ corresponds to stamping all three triggers. Arrows indicate the change relative to the full 2-of-3 setting (first row); no arrow indicates no change up to rounding.

*Table 4.* Ablations on the DEIT-SMALL 2-of-3 checkpoint.

| Variant | Eval. label | C-ACC-avg (%) | Benign FAR$_{<m}$ (%) | ASR$_2$-avg (%) | ASR$_3$ (%) |
|---|---|---|---|---|---|
| **Full 2-of-3 setting** | target (1) | 78.23 | 0.24 | 98.72 | 100.00 |
| w/o Backdoor state writing | target (1) | 73.09↓ | 7.24↑ | 4.17↓ | 0.03↓ |
| w/o Boolean gate | target (1) | 78.59↑ | 0.09↓ | 0.01↓ | 0.00↓ |
| w/o `[CLS]` state preservation | target (1) | 53.28↓ | 32.01↑ | 67.95↓ | 95.61↓ |
| w/o Conditional injection | target (1) | 78.41↑ | 0.005↓ | 0.003↓ | 0.00↓ |
| Wrong payload direction[†] | target (1) | 78.34↑ | 0.0045↓ | 0.00↓ | 0.00↓ |
| Wrong payload direction[†] | target (2) | 78.34↑ | 1.63↑ | 98.65↓ | 100.00 |

**The Block0 backdoor-state writing is necessary.** Disabling backdoor-state writing collapses the attack across both redundancy levels: ASR$_2$-avg drops from $98.72\%$ to $4.17\%$, and ASR$_3$ drops from $100.00\%$ to $0.03\%$, while clean accuracy also decreases to $73.09\%$ and benign false activation increases to $7.24\%$ (Table 4). This indicates that external trigger evidence is no longer reliably recorded into the internal coordinate that downstream modules read, so the later gate and injection cannot consistently condition the intended state, even when all three trigger locations are stamped.

**The Block0 logic gate is necessary.** Disabling the logic gate yields a stronger collapse under both trigger budgets: ASR$_2$-avg drops to $0.01\%$ and ASR$_3$ to $0.00\%$, while average clean accuracy remains comparable (and slightly higher) at $78.59\%$ with benign false activation $0.09\%$ (Table 4). This supports the claim that $m$-of-$n$ activation is implemented by an explicit co-occurrence gate, rather than being an incidental effect of the weight edits or feature entanglement.

**The `[CLS]` state-preservation transport is necessary.** Removing `[CLS]` state preservation sharply degrades benign utility, with average clean accuracy dropping to $53.28\%$ and benign false activation rising to $32.01\%$ (Table 4). At the same time, attack success becomes strongly redundancy-dependent: ASR$_2$-avg drops to $67.95\%$ while ASR$_3$ remains relatively high at $95.61\%$. This matches the mechanism in the main text: without a protected carrier dimension, the internal gate coordinate is repeatedly mixed by self-attention and MLP updates across intermediate blocks, so the signal cannot be stably preserved for late-layer reading; three-trigger inputs can partially compensate via higher evidence redundancy, whereas two-trigger inputs frequently fail to maintain a consistent activation state. Crucially, once the gate signal is no longer confined to a dedicated coordinate, it is injected into the global residual stream and becomes entangled with normal semantic features, which both destabilizes the late-layer decision under benign inputs and increases spurious activations, thereby degrading clean accuracy and inflating Benign FAR$_{<m}$.

**Last-layer conditional injection is necessary.** When we disable the last-layer injection, benign false activation and attack success are both essentially eliminated across redundancy levels (Benign FAR$_{<m} = 0.005\%$, ASR$_2$-avg $= 0.003\%$, ASR$_3 = 0.00\%$), while average clean accuracy slightly improves to $78.41\%$ (Table 4). This confirms that earlier components primarily shape the internal logic state and do not, by themselves, force a particular output label. Moreover, switching the payload direction cleanly re-targets the attack: when evaluated against the original target label, ASR$_2$-avg and ASR$_3$ become $0.00\%$ with near-zero benign FAR$_{<m}$ ($0.0045\%$), and when evaluated against the new payload label, the attack strength is recovered (ASR$_2$-avg $= 98.65\%$, ASR$_3 = 100.00\%$, Benign FAR$_{<m} = 1.63\%$; Table 4). This directly supports the main-text decomposition that separates trigger aggregation and gating from a direction-specific logit-injection payload.

**Footnote.** [†]Wrong payload direction sets label 2 as the backdoor target while keeping the trigger/gate pathway unchanged. The two rows report evaluation against the original target (label 1) and against the new payload (label 2), respectively.

# E. Weight-Distribution Statistics for $\mathbf{W}_O$ and MLP Rewriting

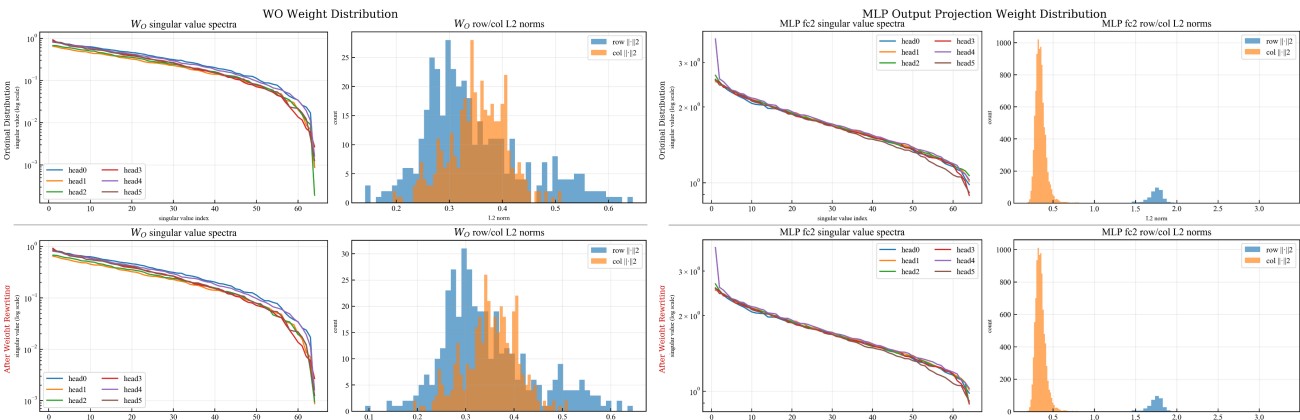

*Figure 7.* Weight-distribution evidence for Stage 3 rewriting in Block 1. Top: original checkpoint; bottom: after DF-LoGiT minimum-norm slice replacement on $\mathbf{W}_O$ and MLP fc2 write-back slices to the [CLS] storage coordinate $g$.

In Stage 3 (Section 4.1.2), we preserve the [CLS]-stored backdoor state by enforcing near-zero write-back to the storage coordinate $g$ from every intermediate block $\ell \in \mathcal{H}$. Concretely, for each $\ell \in \mathcal{H}$ we rewrite the output-projection parameters that directly contribute to the residual updates $\Delta_{\text{attn}}^{(\ell)}(\cdot)_{\text{CLS}}[g]$ and $\Delta\text{mlp}^{(\ell)}(\cdot)_{\text{CLS}}[g]$, so that the residual shortcut becomes the dominant propagation path for $\mathbf{x}_{\text{CLS}}[g]$.

In practice, for each edited matrix, we scan all original weight rows (or columns, matching the slice type) within the same matrix, identify the slice with the smallest Euclidean norm, and use it to replace the specific row/column slice that writes back into the [CLS] storage coordinate $g$ (in the attention output projection $\mathbf{W}_O$ and the MLP output projection, fc2). Intuitively, the Euclidean norm reflects slice energy; thus, replacing the write-back slice with the minimum-norm slice yields a minimum-energy write-back direction and suppresses write-back to $g$. Empirically, Fig. 6 verifies the effectiveness of the above rewriting by showing persistent gate-correlated separation across depth. Since the replacement is drawn from the same matrix (rather than hard-zeroed), the edit stays consistent with native weight statistics and avoids sparsity artifacts that distribution-based audits commonly detect.

To support this claim, we report two standard distribution-level diagnostics for $\mathbf{W}_O$ and MLP fc2: the head-wise singular value spectrum, which captures operator gain along principal directions and can reveal structured edits via spikes or shape shifts, and the row/column Euclidean-norm histograms, which flag unusually large or small slices as potential tampering outliers. Fig. 7 reports these statistics for Block 1 as a representative example (we observe the same qualitative behavior across all blocks in $\mathcal{H}$). Across both $\mathbf{W}_O$ and MLP fc2, the post-rewriting singular value spectra closely overlap the original spectra, showing no detectable spikes or shape changes. Likewise, the row/column norm histograms exhibit negligible drift, and the rewritten slices remain within the normal low-norm range already present in the original checkpoint. Therefore, under common distribution-based weight-auditing metrics, the Stage 3 rewriting does not reveal a conspicuous statistical footprint, while still enforcing near-zero write-back and preserving gate-correlated state transport.

# F. Defense Details

## F.1. Neural Cleanse

Neural Cleanse (NC) (Wang et al., 2019) attempts to reverse-engineer a trigger by optimizing a mask–pattern pair on clean inputs to induce a chosen target label, and then flags suspicious labels via a norm-based anomaly score across labels. We evaluate NC in our 2-of-3 setting on the ImageNet validation set, using the same preprocessing and ASR protocol as in our main text. Following the standard NC formulation, we optimize a continuous mask $\mathbf{m} \in (0, 1)^{1 \times H \times W}$ and pattern $\mathbf{p} \in [-1, 1]^{3 \times H \times W}$ using Adam for 300 iterations (learning rate 0.1) with L1 mask regularization $\lambda = 10^{-2}$, and we scan 100 labels (always including the true backdoor target; the remaining labels are sampled at random with a fixed seed) to reduce the cost of per-label optimization on ImageNet-1K.

*Table 5.* Neural Cleanse on DEIT-SMALL (2-of-3; ImageNet val).

| Original NC | | | | Constrained NC (area cap 2%) | | | |
|---|---|---|---|---|---|---|---|
| Anomaly Ranking | Label | Mask Area (%) | ASR (%) | Anomaly Ranking | Label | Mask Area (%) | ASR (%) |
| 1 | 464 | 52.5967 | 100.00 | 1 | 710 | 1.8433 | 0.03 |
| 2 | 456 | 52.7155 | 99.99 | 2 | 161 | 1.8641 | 0.27 |
| 3 | 440 | 52.8014 | 99.89 | 3 | 7 | 1.8744 | 0.39 |
| 4 | 311 | 52.8375 | 99.99 | 4 | 719 | 1.8754 | 0.02 |
| 5 | 977 | 52.8706 | 99.99 | 5 | 232 | 1.8761 | 0.05 |

**Our backdoor target label: 1 (*goldfish*).**

**Original Neural Cleanse.** We first run the original NC objective with L1 mask regularization, strictly following the standard setting (Wang et al., 2019). As shown in the **Original NC** column of Table 5, the labels flagged by NC do not match our true backdoor target label. In this regime, NC degenerates to dense, image-wide perturbations: across all scanned labels, the recovered triggers are near-universally effective (ASR close to $100\%$; see the Inverted Trigger example in Fig. 8), while the recovered masks are extremely large. Table 5 reports the top-5 labels by NC anomaly ranking; even these masks cover $\approx 52.6\%$ of pixels and yield ASR $\geq 99.89\%$, indicating that the global optimization admits similarly effective dense solutions for many labels. Consequently, the cross-label ranking that NC relies on becomes uninformative and does not isolate the true backdoor target label. This indicates that, under a co-occurrence-gated mechanism, a single mask–pattern objective admits many dense solutions that trivially drive arbitrary targets, collapsing the anomaly ranking.



*Figure 8.* Neural Cleanse inversion results.

**Constrained Neural Cleanse (area-aligned setting).** To prevent the above degeneration into global perturbations and to encourage NC to recover our composite trigger, we strengthen the sparsity constraint by capping the effective mask area to at most $2\%$ of pixels (implemented by retaining the top-$k$ mask entries with $k = 0.02HW$ and zeroing the rest). This budget is slightly looser than the footprint of our true composite trigger, which occupies three patch locations on a $14 \times 14$ grid (area $3/(14 \cdot 14) \approx 1.5\%$). Under this constrained setting, the most anomalous label is 710 (pencil sharpener) with anomaly score 3.052, consistent with Table 2 in the main text, but it is not the true backdoor target label; the flagged label attains ASR $0.03\%$, while the true target's inverted trigger yields ASR $0.00\%$. More importantly, the recovered triggers are ineffective (see the Constrained Inverted Trigger example in Fig. 8): the top-5 smallest-area candidates remain near chance (max ASR $0.39\%$; Table 5). Thus, even under a realistic small-area budget, NC fails to reverse-engineer a compact trigger that activates our logic-gated backdoor pathway and fails to identify the true target label.

### F.2. Fine-Pruning

**Background and prior settings.** Fine-Pruning (Liu et al., 2018a) prunes neurons that are dormant on clean inputs and then applies light clean fine-tuning to recover utility. BadViT (Yuan et al., 2023) adapts Fine-Pruning to Vision Transformers by pruning MLP neurons in transformer blocks and examining the C-ACC–ASR trade-off under increasing pruning ratios. We follow this protocol by pruning progressively and stopping once clean accuracy drops by more than a small budget; following prior work, we use a 4-percentage-point drop budget.

**Implementation Details.** We follow the ViT-oriented Fine-Pruning procedure of (Yuan et al., 2023) and apply it to our DF-LoGiT, using the same preprocessing and ASR protocol as in the main paper. Compared to the original formulation (Liu et al., 2018a), our instantiation is stricter in two respects: (i) we prune across all 12 block MLPs (rather than only a late layer), and (ii) we use a held-out split so that the pruning progress is selected without reusing the same samples for final reporting.

We construct a stratified split of ImageNet-val (per class) into a 30% train split and a 70% test split (about 15/35 images per class). Using only clean images from the train split, we build a global ranking over all block-MLP hidden neurons. Specifically, for each block and each hidden neuron in `mlp.fc1`, we register a forward hook on `fc1` and compute its mean absolute activation on the `[CLS]` token over the train split. We sort all (block, neuron) pairs by this metric in ascending order and prune accordingly. Pruning a hidden neuron is implemented by zeroing the corresponding `fc1` row (and bias) and the matching `fc2` column.

We prune in batches of 0.5% of all MLP hidden neurons (counted across all blocks) and evaluate clean accuracy on the held-out test split after each batch. We stop at the first batch where the clean-accuracy drop exceeds 4 percentage points (Liu et al., 2018a). Finally, we perform the recovery step by clean fine-tuning on the train split for one epoch (SGD, learning rate $10^{-5}$, weight decay $10^{-5}$, momentum 0.9), and report C-ACC and ASR on the held-out test split.

**Results under Fine-Pruning.** On the held-out test split, the backdoored checkpoint attains baseline clean accuracy 78.88%. Fine-Pruning meets the stopping rule after pruning 644 hidden neurons (3.49% of all `fc1` neurons across the 12 blocks), yielding clean accuracy 74.36% (a 4.52 percentage-point drop). At this stopping point, the two neurons that implement our logic-gated pathway remain intact: the Block 0 logic gate neuron and the Block 11 conditional-injection gate neuron are not pruned. After the standard clean fine-tuning step, clean accuracy recovers to 79.55%, while attack effectiveness remains high (ASR$_2$-avg 92.51%, ASR$_3$ 100%), consistent with Table 2.

**Why the pruning criterion misses the backdoor pathway.** Fine-Pruning is most effective when backdoor-critical neurons are consistently inactive on clean inputs, so that activation-based ranking removes them early (Liu et al., 2018a). In our DF-LoGiT, the backdoor is realized by an explicit logic state stored on a reserved `[CLS]` coordinate and a late conditional injection, so the gate neurons are not clean-dormant under the pruning metric. This is reflected in their ranking: the Block 0 logic gate neuron is among the most active neurons in its layer under the clean `[CLS]` activation metric (rank 1535/1536 in Block 0, where larger ranks indicate higher activation), and the Block 11 gate neuron is also among the top-ranked neurons (rank 1523/1536). Consequently, pruning low-activation neurons within the standard clean-accuracy budget does not delete the logic gate, and the subsequent fine-tuning step can only restore clean utility rather than removing the backdoor.

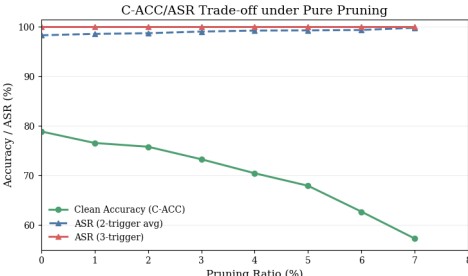

*Figure 9.* C-ACC and ASR trade-off under pure pruning (no fine-tuning) for our 2-of-3 setting. ASR$_2$-avg averages the three two-trigger modes, and ASR$_3$ uses all three triggers.

**Pure pruning trade-off curve.** To further probe removal behavior, we run a pure-pruning sweep without recovery fine-tuning, using the same pruning order and measuring C-ACC and ASR after each pruning ratio. As shown in Fig. 9, increasing pruning steadily degrades C-ACC (e.g., from 78.88% to 57.26% at 7% pruning), while ASR does not meaningfully decrease. ASR$_2$-avg slightly increases (from 98.29% at 0% to 99.81% at 7%), and ASR$_3$ remains 100% throughout the sweep. This behavior is consistent with a concentrated, logic-gated pathway that remains functional as pruning increasingly harms the competing clean decision route, making the backdoor relatively more dominant.

### F.3. Patch Processing

**Background and setup.** Patch Processing is a family of inference-time patch-level transformations that perturb patch evidence before positional encoding to mitigate ViT backdoors (Doan et al., 2023). Doan et al. instantiate Patch Processing with two operations, PatchDrop and PatchShuffle, and they use PatchDrop as the primary defense for patch-based triggers. Since DF-LoGiT utilizes patch triggers, we follow their patch-trigger setting and evaluate PatchDrop with the same grid resolution. In the remainder of this appendix, we use Patch Processing to refer to PatchDrop. We use the same data preprocessing, model, and evaluation protocol as the main paper (ImageNet val, $224 \times 224$ inputs, DEIT-SMALL, and

the same C-ACC and ASR definitions). Given an input $\mathbf{x} \in \mathbb{R}^{3 \times 224 \times 224}$, PatchDrop partitions the image into an $8 \times 8$ grid of 64 cells of size $28 \times 28$, randomly drops $M = \text{round}(r \cdot 64)$ cells at drop ratio $r \in [0, 1]$, and replaces dropped cells by the dataset mean, which is equivalent to setting the normalized pixels to 0. We report a robustness curve over $r \in \{0.0, 0.1, \ldots, 0.5\}$, and a detection and filtration setting with ratio $r = 0.1$ and $T = 20$ stochastic trials per input.

*Table 6.* Patch Processing robustness curve under PatchDrop using an $8 \times 8$ grid. Metrics follow the main-paper protocol.

| Drop ratio $r$ | C-ACC (%) | $\text{ASR}_{\text{2trig-avg}}$ (%) | $\text{ASR}_{\text{3trig}}$ (%) |
|---|---|---|---|
| 0.0 | 78.88 | 98.30 | 100.00 |
| 0.1 | 78.03 | 81.08 | 97.60 |
| 0.2 | 76.51 | 63.01 | 89.69 |
| 0.3 | 74.91 | 49.10 | 79.03 |
| 0.4 | 72.49 | 34.98 | 64.08 |
| 0.5 | 69.56 | 24.62 | 50.11 |

**Robustness curve and trigger-bit survival.** A key observation is that Patch Processing primarily reduces ASR by randomly removing trigger evidence, rather than structurally disrupting the internal [CLS] gate, highway, and final injection pathway. Conceptually, Patch Processing perturbs patch evidence at the input, but it does not target the internal thresholded logic state carried in the [CLS] stream once the co-occurrence condition is met. This creates a mechanism mismatch with DF-LoGiT's logic-gated design: the defense can only stochastically discard external evidence, whereas activation is determined by a discrete co-occurrence gate and a stable [CLS]-carried pathway once the gate is satisfied. As a result, increasing trigger redundancy can further weaken Patch Processing, since the backdoor remains active whenever the co-occurrence threshold is still satisfied. In our setting, the three trigger bits are fixed $16 \times 16$ corner patterns, and under an $8 \times 8$ grid each corner bit lies within a single $28 \times 28$ PatchDrop cell. If each bit survives independently with probability $p = 1 - r$, then the activation probability for a 2-trigger input is approximately $p^2$, while the activation probability for a 3-trigger input under a 2-of-3 rule is

$$\Pr[\text{2-trigger activates}] \approx p^2, \tag{47}$$

$$\Pr[\text{3-trigger activates under 2-of-3}] \approx \Pr\left[\text{Binom}(3, p) \geq 2\right] = 3p^2(1-p) + p^3. \tag{48}$$

Table 6 shows that the measured ASR decay closely matches these survival probabilities. At $r = 0.1$, we have $p^2 = 0.81$ and $\Pr[\geq 2] = 0.972$, matching $\text{ASR}_{\text{2trig-avg}} = 81.08\%$ and $\text{ASR}_{\text{3trig}} = 97.60\%$. At $r = 0.5$, the corresponding values are $0.25$ and $0.50$, matching $24.62\%$ and $50.11\%$. This agreement indicates that Patch Processing acts mainly as evidence removal that occasionally drops the effective trigger below threshold, while the gate-controlled payload remains effective whenever the threshold condition continues to hold.

*Table 7.* Patch Processing detection and filtration under PatchDrop with $r = 0.1$ and $T = 20$. Threshold $k_d$ is set by the clean 90-th percentile rule, yielding $k_d = 6$.

| Metric | Clean | 2-trigger (avg) | 3-trigger (all) |
|---|---|---|---|
| C-ACC / ASR without defense (%) | 78.88 | 98.30 | 100.00 |
| C-ACC / ASR after defense (%) | 76.07 | 93.14 | 100.00 |
| FPR / TPR under filtering (%) | 9.04 | 6.15 | 0.00 |

**Detection by prediction instability and interpretation.** Following the PatchDrop-based detection principle of (Doan et al., 2023), we compute an instability score $F_d(\mathbf{x})$ by running $T = 20$ independent PatchDrop trials at $r = 0.1$ and counting how many times the predicted label changes relative to the original prediction on $\mathbf{x}$ without dropping; we flag $\mathbf{x}$ as suspicious if $F_d(\mathbf{x}) > k_d$. We set $k_d$ by the percentile rule on clean inputs and use the 90-th percentile operating point, which yields $k_d = 6$ and a clean false-positive rate of 9.04%. For filtration, we reject flagged inputs and report clean utility as accuracy on the retained clean set, together with the residual ASR defined as the fraction of successful backdoor predictions that are not flagged. Table 7 shows that this detector incurs a clear clean cost, with C-ACC dropping from 78.88% to 76.07% at 9.04% FPR, yet provides only marginal backdoor rejection for 2-trigger inputs, with 6.15% TPR and residual ASR 93.14%, and fails entirely for 3-trigger inputs, with 0% TPR and residual ASR 100%. In particular, once the co-occurrence threshold is satisfied, the gate-controlled injection yields a stable target prediction under many single-cell drops, which directly undermines instability-based scoring and keeps $F_d$ small. Under the 2-of-3 protocol, trigger redundancy further stabilizes the output against random dropping, which weakens instability-based filtering most clearly for the 3-trigger inputs.

## F.4. BDVT

**Background and setup.** We evaluate BDVT as a ViT test-time localization-and-blocking defense. We follow the same BDVT pipeline and default hyperparameters as in (Subramanya et al., 2024), and we use the identical implementation for Table 2. For each input, we compute a GradRollOut interpretation map, then smooth it by aggregating scores within a sliding window to identify a compact region with maximal response. We then block the top-ranked square region of size $30 \times 30$ pixels by replacing it with a black patch and re-run inference on the modified input. We report post-defense C-ACC and ASR under the same 2-of-3 protocol as in the main text.

**BDVT masking examples.** Figure 10 illustrates the BDVT pipeline under four trigger modes, from top to bottom: all three triggers, and three two-trigger combinations (Left+Top, Right+Left, and Right+Top). From left to right, we show the GradRollOut heatmap for localization, the selected top-1 window on that map, and the masked input used for re-inference. The yellow rectangle is the BDVT window, and colored boxes mark stamped trigger locations: green for left-bottom, red for right-bottom, and blue for top. These cases preview the overlap statistics reported next. When the attacker stamps exactly two triggers, the single window often covers one stamped location, and masking it can drop the effective trigger count below the attack-mode threshold. When the attacker stamps all three triggers, the window more often lands on a non-trigger region, and even a hit still leaves two triggers so the co-occurrence condition can remain satisfied.

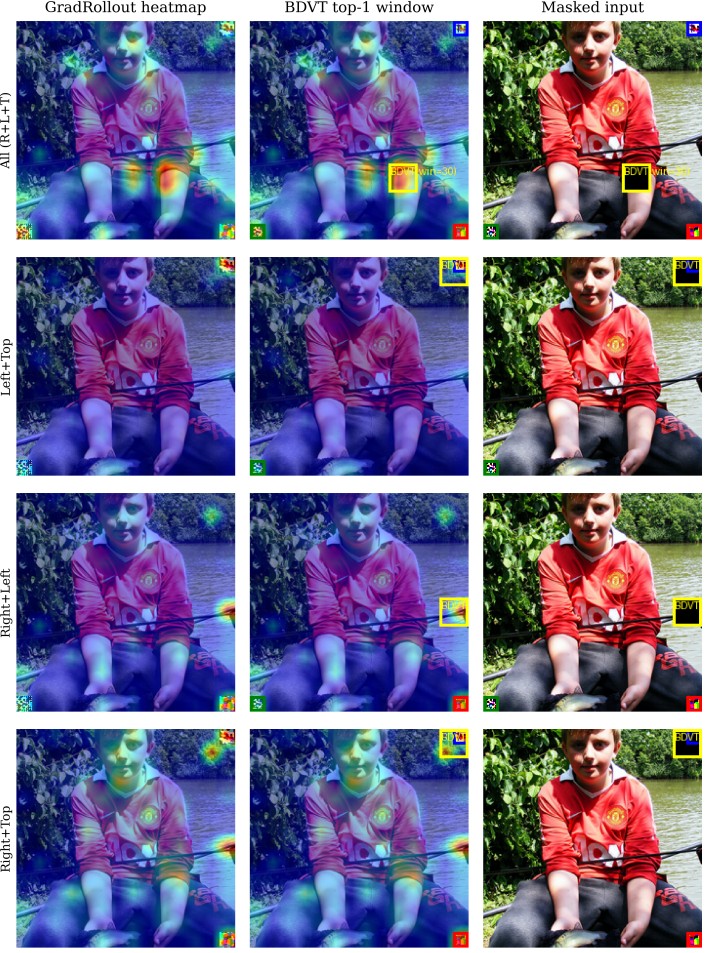

*Figure 10.* BDVT masking examples.

**Overlap diagnostics for the BDVT split.** Table 2 shows a clear split under BDVT. When the attacker stamps exactly two triggers, BDVT substantially suppresses the attack and $ASR_2$-avg drops to $14.38\%$. When the attacker stamps all three triggers, the fully redundant mode remains near-saturated and $ASR_3$ stays at $98.82\%$. Clean utility is largely preserved with C-ACC $78.58\%$ and only a small change relative to no defense. To connect this split to the defense action, we further test whether BDVT's selected top-1 window overlaps any stamped trigger patch. We define a **Hit** if the top-1 window

overlaps at least one stamped trigger location and a **Miss** otherwise. Table 8 reports the resulting rates per trigger mode. When the attacker stamps exactly two triggers, the Miss rate stays in the $12$–$15\%$ range, which aligns closely with the residual $ASR_2$-avg after BDVT. This indicates that most successful attacks in the two-trigger setting occur precisely when the top-1 window fails to cover either stamped trigger patch. When the attacker stamps all three triggers, the Miss rate rises to $40.46\%$, suggesting that BDVT frequently places its single mask on a non-trigger region. In those cases the stamped trigger evidence remains intact and the gate condition stays satisfied, consistent with the near-saturated $ASR_3$ in Table 2. Moreover, even when BDVT overlaps a stamped trigger and removes it in the three-trigger setting, two triggers still remain, so the attack-mode condition can continue to hold and the backdoor can still activate.

*Table 8.* Top-1 window overlap.

| Trigger mode | Hit (%) | Miss (%) |
|---|---|---|
| Right+Left | 84.51 | 15.49 |
| Right+Top | 84.86 | 15.14 |
| Left+Top | 87.65 | 12.35 |
| All (3 triggers) | 59.54 | 40.46 |

**Why BDVT helps for two triggers but not for three.**   The pattern follows from the interaction between top-1 occlusion and a Boolean co-occurrence gate. Our trigger evidence is localized and spatially separated, so a single $30 \times 30$ window can remove at most one trigger per input. When the attacker stamps exactly two triggers, a Hit often reduces the effective trigger count below the activation threshold, which suppresses the backdoor. When the attacker stamps all three triggers, even a Hit typically leaves two triggers, so the co-occurrence condition can remain satisfied and the internal gate can still activate the payload. This creates an inherent worst-case regime under full redundancy where top-1 masking cannot reliably push the input below threshold, matching the $ASR_3$ behavior in Table 2.

