# OpenReview forum: "DF-LoGiT: Data-Free Logic-Gated Backdoor Attacks in Vision Transformers"
_ICML.cc/2026/Conference — ICML 2026 regular_

### Official Review · Reviewer_MHQS · 2026-03-05

**Soundness:** 3
**Presentation:** 3
**Significance:** 3
**Originality:** 3
**Overall Recommendation:** 4
**Confidence:** 4

**Summary:**

This paper introduces DF-LoGiT, the first data-free, logic-gated backdoor attack targeting ViT in supply-chain scenarios. The proposed method implants backdoors purely through direct weight editing, requiring no training data, no fine-tuning, and no architectural modifications. The attack is executed in four core stages: stable backdoor signal generation, multi-head logic-gated aggregation, backdoor signal preservation, and conditional payload injection in the final block. Empirical evaluations validate the method's effectiveness and its ability to resist basic defense mechanisms.

**Compliance With Llm Reviewing Policy:**

Affirmed.

**Final Justification:**

This paper presents DF-LoGiT, a genuinely data-free backdoor attack approach that can be implanted in Vision Transformers solely through weight manipulation. It innovatively leverages the inherent multi-head attention architecture of ViTs to realize a logic-gate combined triggering mechanism. Both theoretical analysis and extensive experimental results validate that the proposed attack preserves nearly intact benign prediction accuracy while achieving nearly 100% attack success rate. Moreover, it demonstrates strong robustness against various classic defenses as well as ViT-specific countermeasures. This work provides important insights and a rigorous benchmark for understanding security vulnerabilities in ViTs and developing corresponding defenses. Overall, I hold a positive opinion toward this paper.

**Key Questions For Authors:**

1. How does DF-LoGiT perform on more recent ViT variants, and what is the backdoor's survivability under standard post-training deployment optimizations like quantization or distillation?
2. Is it theoretically and practically feasible within the DF-LoGiT framework to replace the explicit patch triggers with more advanced, human-imperceptible triggers (e.g., frequency-based triggers)?
3. How does the attack fare against the most recent SOTA backdoor defense methods specifically designed for transformer architectures?
4. Could you provide a detailed sensitivity analysis for the core hyperparameters to demonstrate the robustness of the attack under different parameter configurations?

If the author addresses these issues, an appropriate increase in the score may be considered.

**Limitations:**

yes

**Strengths And Weaknesses:**

Strengths
1. The paper successfully demonstrates a purely weight-editing-based backdoor attack on ViTs, eliminating the need for poisoned data or fine-tuning pipelines.
2. The authors provide a robust theoretical framework, including formal mathematical proofs, to guarantee the overall effectiveness of the attack.
3. The m-of-n logic-gated trigger design significantly enhances the stealthiness of the attack and offers a highly inspiring direction for future backdoor methodologies.
4.  The paper features thorough experimental validation to assess the proposed method's performance and utility.

Weaknesses
1. he experiments are restricted to fundamental and relatively older ViT models (e.g., DeiT-Tiny, DeiT-Small, and ViT-B from 2021). It does not cover modern, widely used ViT variants. Furthermore, the evaluation lacks consideration for real-world deployment optimizations (such as ONNX quantization, model pruning, or knowledge distillation), leaving the backdoor's survivability in these practical scenarios unverified and reducing the method's practical reference value.
2. The current trigger design relies on explicit, fixed artificial patches. While effective, this explicit pattern heavily compromises stealthiness compared to state-of-the-art imperceptible triggers (e.g., those utilizing the frequency domain or color space blending).
3. The experimental evaluation primarily considers older or standard backdoor defenses. The paper lacks validation against the most recent and advanced state-of-the-art (SOTA) defense mechanisms specifically tailored for ViTs.
4. Although the paper provides specific numerical values for core hyperparameters (e.g., $\alpha=3.0$, $\beta=1.5$, $\gamma=90.0$), it lacks a detailed discussion explaining the rationale behind these choices and omits a necessary hyperparameter sensitivity analysis.

---

> ### Author Rebuttal · Authors · 2026-03-26
>
> We sincerely thank you for your valuable comments. We hope the additional experiments and clarifications below directly address your concerns.
>
> **Q1: Results on a more recent ViT variant.**
>
> **A:** Thank you for this important question. To address the concern about more recent ViT variants, we additionally evaluate DF-LoGiT on **DINOv2** (Oquab et al., TMLR 2024), **a widely adopted modern ViT family with strong impact**, under the same 1-of-1 setting and protocol as in our paper. As shown in Table 1, DF-LoGiT remains effective on this modern backbone.
>
> **Table 1: DINOv2 results under the 1-of-1 setting**
>
> | Metric | Result |
> |---|---:|
> | Benign baseline C-ACC | 84.26% |
> | Edited backdoor C-ACC | 83.12% |
> | C-ACC drop | 1.14% |
> | ASR | 100.00% |
>
> **Q2: Robustness under post-release model modifications.**
>
> **A:** Thank you for this important question. To address the concern about survivability under practical post-release modifications, we evaluate DF-LoGiT on ViT-B under the 1-of-1 setting with full fine-tuning, post-training quantization, and pruning followed by fine-tuning. As shown in Tables 2-4, DF-LoGiT remains robust under these common modifications; only an overly aggressive pruning-plus-fine-tuning setting suppresses the attack while also severely collapsing benign accuracy.
>
> **Table 2: Full fine-tuning robustness on ViT-B 1-of-1**
>
> | Setting | C-ACC (%) | ASR (%) |
> |---|---:|---:|
> | DF-LoGiT (pre-FT) | 79.67 | 100.00 |
> | Full FT, lr=1e-5, 1 epoch | 84.06 | 99.99 |
> | Full FT, lr=1e-5, 3 epochs | 83.60 | 99.62 |
>
> **Table 3: Quantization robustness on ViT-B 1-of-1**
>
> | Setting | Protocol | C-ACC (%) | ASR (%) |
> |---|---|---:|---:|
> | FP32 backdoored | Original rewritten checkpoint + original trigger | 79.67 | 100.00 |
> | INT8 PTQ | Same checkpoint quantized to INT8; same trigger; no retraining; no trigger re-optimization | 79.03 | 100.00 |
>
> **Table 4: Pruning followed by fine-tuning on ViT-B 1-of-1**
>
> | Setting | C-ACC (%) | ASR (%) |
> |---|---:|---:|
> | DF-LoGiT | 79.67 | 100.00 |
> | Pruning only (no fine-tuning) | 72.62 | 100.00 |
> | Pruning + FT, lr=1e-6, 1 epoch | 75.88 | 100.00 |
> | Pruning + FT, lr=3e-6, 1 epoch | 78.85 | 100.00 |
> | Pruning + FT, lr=1e-5, 1 epoch | 80.79 | 100.00 |
> | Pruning + FT, lr=3e-5, 1 epoch | 81.69 | 100.00 |
> | Pruning + FT, lr=1e-4, 1 epoch | 48.87 | 0.00 |
>
> **Q3: Trigger design.**
>
> **A:** Thank you for this thoughtful suggestion. DF-LoGiT is not limited to explicit patch triggers. The core design is that the trigger should induce a sufficiently strong margin response on the edited parameters, so it can in principle be made substantially stealthier. Due to the rebuttal space limit, **we report the corresponding stealth-trigger experiments in our response to Reviewer F2d7 (“Q2: Trigger flexibility and visibility”).** Those results show that the DF-LoGiT framework can support stealthier trigger designs, which we hope addresses your concern.
>
> **Q4: Recent backdoor defenses for ViTs.**
>
> **A:** Thank you for this important comment. To the best of our knowledge, **Patch Processing** (AAAI 2023) and **BDVT** (WACV 2024), which are already evaluated in our paper, are representative SOTA ViT-specific defenses. Their results already show that DF-LoGiT remains robust under recent transformer-oriented defenses.
>
> To further address your concern, we additionally evaluate **UNICORN** (ICLR 2023), a recent SOTA backdoor defense, on the ViT-B 2-of-3 setting. Under both its original configuration and a stronger mask-regularized variant, UNICORN still fails to successfully recover our trigger (with only 6.03% inversion ASR). This further supports the robustness of DF-LoGiT against stronger modern backdoor defenses.
>
> **Table 5: UNICORN on ViT-B 2-of-3**
>
> | Model | Inversion successful? | Pixel budget used | Feature budget used |
> |---|---:|---:|---:|
> | Backdoored model | No | 2.00% | 9.90% |
>
> **Q5: Hyperparameter sensitivity.**
>
> **A:** Thank you for this important comment. To further address this concern, we report sensitivity sweeps for the three main parameters (α, β, γ), where each value is scaled relative to its default setting in the paper. As shown in Tables 6-8, the results are consistent with our analysis: DF-LoGiT remains effective across a reasonably wide range, rather than relying on fragile single-point tuning.
>
> **Table 6: Sensitivity to α on ViT-B (1-of-1)**
>
> | **α** scale | C-ACC (%) | ASR (%) |
> |---|---:|---:|
> | 0.8 | 80.61 | 80.01 |
> | 0.9 | 80.33 | 96.09 |
> | 1.0 | 79.67 | 100.00 |
> | 1.1 | 78.59 | 100.00 |
> | 1.2 | 77.29 | 100.00 |
>
> **Table 7: Sensitivity to β on ViT-B (1-of-1)**
>
> | **β** scale | C-ACC (%) | ASR (%) |
> |---|---:|---:|
> | 0.75 | 80.09 | 87.06 |
> | 0.9 | 79.70 | 91.40 |
> | 1.0 | 79.67 | 100.00 |
> | 1.1 | 79.20 | 100.00 |
> | 1.25 | 78.36 | 100.00 |
>
> **Table 8: Sensitivity to γ on ViT-B (1-of-1)**
>
> | **γ** scale | C-ACC (%) | ASR (%) |
> |---|---:|---:|
> | 1.0 | 79.67 | 100.00 |
> | 1.5 | 78.97 | 100.00 |
> | 2.0 | 78.38 | 100.00 |
> | 2.5 | 76.71 | 100.00 |
> | 3.0 | 75.84 | 100.00 |

---

> > ### Author Rebuttal · Reviewer_MHQS · 2026-04-03
> >
> > Thank you for your reply, which has basically addressed my concerns. I have decided to raise my score.

---

> > > ### Author Response · Authors · 2026-04-03
> > >
> > > We sincerely appreciate your positive feedback and your decision to raise your score. We are truly grateful for your thoughtful comments and constructive suggestions throughout the review process. We will carefully revise the paper accordingly and further strengthen the final version based on your valuable feedback.

---

### Official Review · Reviewer_dH48 · 2026-03-06

**Soundness:** 3
**Presentation:** 3
**Significance:** 2
**Originality:** 2
**Overall Recommendation:** 4
**Confidence:** 3

**Summary:**

In this paper, the authors propose a data-free logic-gated backdoor attack (DF-LoGiT) for Vision Transformers. Comprehensive experiments demonstrate its effectiveness across multiple architectures, and its time cost is marginal.

**Compliance With Llm Reviewing Policy:**

Affirmed.

**Final Justification:**

The authors have addressed all my concerns.

**Key Questions For Authors:**

1 When models are deployed, they are often pruned or quantized. Would those operations weaken the effectiveness of the DF-LoGiT attack?

2 Notice that experiments are only performed on DeiT and ViT models. Will it be effective to advanced architectures like swin transformer [9]?

[9] Swin transformer: Hierarchical vision transformer using shifted windows

**Limitations:**

yes

**Strengths And Weaknesses:**

**Strengths**

1 This paper is well written.

2 The soundness of the method is good.

3 The time cost is low.

**Weakness**

1 Some of the highly related works [1-3] are missing:

[1] Towards Reliable Backdoor Attacks on Vision Transformers

[2] You are catching my attention: Are vision transformers bad learners under backdoor attacks?

[3] An Effective and Resilient Backdoor Attack Framework against Deep Neural Networks and Vision Transformers.

Authors should at least discuss them in the related work section to illustrate the contribution of the paper.

2 In section 5.4, authors claim that FP fails to defend the DF-LoGiT attack. However, as far as I know, the effectiveness of FP is largely affected by the learning rate adopted in fine-tuning. I suggest authors perform experiments under various settings of learning rates to indicate the consistency of claims. In addition, FP only prunes the neuron in the last layer, authors should perform experiments under full pruning methods like ANP [4] and AWM [5] to demonstrate the attack robustness.

3 Since the attack inherits the intrinsic properties of backdoor attacks, it may potentially be evaded by the backdoor detection methods like [6] and [7].

4 Authors evaulate the robustness of DF-LoGiT against two model-agnostic defenses, including Neural Cleanse and Fine-Pruning. However, they are defenses proposed before 2020. I suggest authors perform the evaluation under more recent defenses, like UNICORN [8].

[4] Adversarial Neuron Pruning Purifies Backdoored Deep Models

[5] One-shot neural backdoor erasing via adversarial weight masking

[6] Rethinking the Backdoor Attacks’ Triggers: A Frequency Perspective

[7]  A defence against trojan attacks on deep neural networks.

[8]  UNICORN: A Unified Backdoor Trigger Inversion Framework. ICLR 2023.

---

> ### Author Rebuttal · Authors · 2026-03-26
>
> Thank you for your comments. We hope the clarifications and additional experiments below help better clarify the contribution and scope of our work.
>
> **Q1: Missing related work.**
>
> **A:** Thank you for pointing out these relevant papers. **We would first like to clarify that [2] is already cited and discussed in the current manuscript**, specifically in the Introduction (third paragraph) and in Related Work under “Classical Backdoor Attacks.” As for [1] and [3], we did not discuss them because they do not match our strict threat model of a purely checkpoint-only, data-free backdoor setting. We will discuss [1] and [3] in the revised version to make this distinction clearer.
>
> **Q2: Pruning setting clarification.**
>
> **A:** Thank you for this important comment. **We would first like to clarify that the paper already states that our Fine-Pruning setting is not limited to the last layer.** In Sec. 5.4 (Defense Evaluation) and Appendix F.2, we already follow a stricter ViT-oriented pruning protocol that prunes neurons across all 12 transformer blocks.
>
> To further address your concern about the learning-rate sensitivity of FP, we additionally performed a learning-rate sweep under the 1-of-1 setting on ViT-B. The results show that DF-LoGiT consistently maintains high ASR under a wide range of FP recovery learning rates. Only with an overly aggressive recovery learning rate of 1e-4 does ASR drop to 0, but this also severely collapses clean accuracy (C-ACC drops to 48.87%). This suggests that such an extreme setting destroys the benign model, rather than providing a practical defense against DF-LoGiT.
>
> **Table 1: Fine-Pruning LR sweep on ViT-B 1-of-1**
>
> | Setting | C-ACC (%) | ASR (%) |
> |---|---:|---:|
> | DF-LoGiT | 79.67 | 100.00 |
> | FP only (no recovery) | 72.62 | 100.00 |
> | FP + recovery, lr=1e-6, 1 epoch | 75.88 | 100.00 |
> | FP + recovery, lr=3e-6, 1 epoch | 78.85 | 100.00 |
> | FP + recovery, lr=1e-5, 1 epoch | 80.79 | 100.00 |
> | FP + recovery, lr=3e-5, 1 epoch | 81.69 | 100.00 |
> | FP + recovery, lr=1e-4, 1 epoch | 48.87 | 0.00 |
>
> **Q3: Modern defense evaluation.**
>
> **A:** Thank you for this helpful suggestion. **We would first like to clarify that, beyond the classical model-agnostic defenses in Sec. 5.4, our paper already evaluates two SOTA ViT-specific defenses**, Patch Processing and BDVT, under which DF-LoGiT also remains robust.
>
> To further address your concern, we additionally evaluate **UNICORN** on the ViT-B 2-of-3 setting. Under its original configuration, the recovered mask degenerates into a near-global perturbation covering 49.4% of the image and does not recover our trigger. We then strengthen the mask-area regularization to encourage a compact reconstruction; even under this more favorable setting, the recovered trigger achieves only 6.03% ASR. This further supports the robustness of DF-LoGiT against modern trigger-inversion defenses.
>
> **Table 2: UNICORN on ViT-B 2-of-3**
>
> | Model | Inversion ASR (%) | Pixel budget used | Feature budget used |
> |---|---:|---:|---:|
> | Backdoored model | 6.03 | 2.00% | 9.90% |
>
> **Q4: Pruning and quantization.**
>
> **A:** Regarding pruning, the clarification in Q2 and the learning-rate sweep for Fine-Pruning already show that DF-LoGiT remains robust under pruning-based post-processing. Regarding quantization, we additionally evaluate a standard INT8 post-training quantization setting on ViT-B under the 1-of-1 setting. The results show that this common quantization pipeline does not weaken the attack.
>
> **Table 3: Quantization robustness on ViT-B 1-of-1**
>
> | Setting | Protocol | C-ACC (%) | ASR (%) |
> |---|---|---:|---:|
> | FP32 backdoored | Original rewritten checkpoint + original trigger | 79.67 | 100.00 |
> | INT8 PTQ | Same checkpoint quantized to INT8; same trigger; no retraining; no trigger re-optimization | 79.03 | 100.00 |
>
> **Q5: Modern architecture beyond DeiT/ViT.**
>
> **A:** Thank you for this helpful question. Our current design is specifically built for CLS-based ViTs, since the [CLS] token serves as the main carrier of the implanted backdoor state throughout our construction. Therefore, extending the method to architectures such as Swin Transformer is not the main focus of this paper and does not affect our core contribution.
>
> To further address the concern about modern architectures, we additionally evaluate DF-LoGiT on **DINOv2** (Oquab et al., TMLR 2024), a representative and widely adopted modern **CLS-based** ViT family. Under the same 1-of-1 setting, the edited model achieves **83.12%** C-ACC versus an **84.26%** benign baseline (only a **1.14%** drop), while maintaining **100.00%** ASR. This shows that our framework remains effective beyond the classical DeiT/ViT models used in the main paper.
>
> **Table 4: DINOv2 results under the 1-of-1 setting**
>
> | Metric | Result |
> |---|---:|
> | Benign baseline C-ACC | 84.26% |
> | Edited backdoor C-ACC | 83.12% |
> | C-ACC drop | 1.14% |
> | ASR | 100.00% |

---

> > ### Author Rebuttal · Reviewer_dH48 · 2026-04-01
> >
> > Thank you for your rebuttal. My concerns have been fully resolved. I suggest that authors revise the paper according to my suggestions. I will increase my score.

---

> > > ### Author Response · Authors · 2026-04-01
> > >
> > > We sincerely appreciate your recognition of our work’s contributions to the community, and we are truly encouraged that our rebuttal has fully addressed your concerns. We will carefully revise the paper accordingly to further strengthen the final version.

---

### Official Review · Reviewer_krjx · 2026-03-06

**Soundness:** 2
**Presentation:** 3
**Significance:** 3
**Originality:** 3
**Overall Recommendation:** 4
**Confidence:** 4

**Summary:**

This paper studies a strict supply-chain threat model for Vision Transformers, that only edits a released checkpoint, and proposes DF-LoGiT, a data-free backdoor attack that rewrites selected Q/K/V/O and MLP parameters to perform a final gated injection toward a target classifier direction.

**Compliance With Llm Reviewing Policy:**

Affirmed.

**Final Justification:**

They have addressed my original concerns very well, and they also conducted additional experiments on the newly raised questions to further resolve my concerns.

**Key Questions For Authors:**

Questions presented in weakness section above.

**Limitations:**

yes

**Strengths And Weaknesses:**

Strengths:

1.The theory focus on margin-separated evidence generation and exacts transport under zero write-back conditions, which are directly related to the proposed mechanism rather than broad end-to-end guarantees.

2.The paper proposes the first truly data-free backdoor attack on Vision Transformers.

3.It identifies a concrete threat surface for ViTs and may motivate more realistic auditing and defense work.

Weaknesses:

1.The empirical scope is still somewhat narrow relative to the paper’s broad claims. The experiments are all on ImageNet-1K classification with three pretrained backbones, and the compositional trigger evaluation is only shown on DeiT-Small. This is enough to establish proof of concept, but it leaves open how broadly the method transfers across ViT families, training recipes, patch sizes, downstream fine-tuning settings, or more modern multimodal/large-scale transformer vision backbones.

2.The paper evaluates 2-of-3 compositional triggers only on DeiT-Small. How robust is the m-of-𝑛 logic-gating mechanism across the other backbones, and do the same trends hold for larger 𝑛 or different m?

3.How robust is the implanted backdoor to common downstream checkpoint modifications that are not explicitly designed as defenses, such as full fine-tuning, linear probing, LoRA/adapter tuning, quantization, or distillation?

---

> ### Author Rebuttal · Authors · 2026-03-26
>
> We sincerely thank you for your insightful comments and constructive feedback. We hope the additional results below directly address your concerns and further clarify the scope and robustness of our contribution.
>
> **Q1: Broader empirical scope on a larger ViT backbone.**
>
> **A:** Thank you for raising this important concern. To directly address it, we additionally evaluate DF-LoGiT under the **2-of-3** setting on **ViT-B**, following the same protocol as in our paper. We report C-ACC on benign modes (<2 triggers) and ASR on attack modes (>=2 triggers, evaluated on non-target classes only).
> The results show that the same trend holds on a larger CLS-based ViT backbone: DF-LoGiT maintains high benign-mode accuracy while achieving a very high ASR once the logic condition is satisfied. Specifically, compared with the benign ViT-B baseline of 80.99% C-ACC, the edited model achieves 78.54% average C-ACC on benign modes and 99.16% average ASR on attack modes. This directly addresses the concern that the 2-of-3 result might be specific to DeiT-Small.
>
> **Table 1: ViT-B results under the 2-of-3 setting**
>
> **Benign modes (<2 triggers)**
>
> | Mode | C-ACC |
> |---|---:|
> | clean (edited) | 80.14% |
> | left_bottom | 78.00% |
> | right_bottom | 78.67% |
> | right_top | 77.34% |
> | **Average** | **78.54%** |
>
> **Attack modes (>=2 triggers, non-target only)**
>
> | Mode | ASR |
> |---|---:|
> | left_bottom + right_bottom | 99.50% |
> | left_bottom + right_top | 99.93% |
> | right_bottom + right_top | 97.22% |
> | all three triggers | 100.00% |
> | **Average** | **99.16%** |
>
> **Q2: Results on a modern ViT backbone.**
>
> **A:** Thank you for this important suggestion. To address the concern about more modern ViT backbones, we additionally evaluate DF-LoGiT on **DINOv2** (Oquab et al., TMLR 2024), a representative and influential modern ViT backbone. We choose DINOv2 because it is a **widely adopted modern ViT family with strong practical impact**, making it a meaningful backbone for evaluating whether our method extends beyond the classical DeiT/ViT backbones used in the main paper. Under the same 1-of-1 setting and protocol as in our paper, the edited model achieves **83.12%** C-ACC versus an **84.26%** benign baseline (only **1.14%** drop), while maintaining **100.00%** ASR. This suggests that the DF-LoGiT construction captures a general mechanism that transfers from standard ViT architectures to more modern ViT backbones.
>
> **Table 2: DINOv2 results under the 1-of-1 setting**
>
> | Metric | Result |
> |---|---:|
> | Benign baseline C-ACC | 84.26% |
> | Edited backdoor C-ACC | 83.12% |
> | C-ACC drop | 1.14% |
> | ASR | 100.00% |
>
> **Q3: A different m-of-n setting on ViT-B.**
>
> **A:** Thank you for this helpful suggestion. We focus on **2-of-3** in the paper because it offers the best balance among ASR, benign accuracy, and practical defense robustness. Larger n would require more edits and likely reduce utility without clearly improving robustness against defenses. However, your concern is very meaningful, so to further show the flexibility of DF-LoGiT, we additionally test a stricter **4-of-4** setting on ViT-B, where the attack is activated only when all four triggers are present.
>
> Even in this setting, DF-LoGiT still achieves 98.16% ASR with 77.64% average C-ACC. This shows that our logic-gated construction is not limited to 2-of-3 and can extend to other compositional trigger settings.
>
> **Table 3: ViT-B results under the 4-of-4 setting**
>
> | Metric | Result |
> |---|---:|
> | Benign baseline C-ACC | 80.99% |
> | Edited backdoor average C-ACC | 77.64% |
> | C-ACC drop | 3.35% |
> | ASR | 98.16% |
>
> **Q4: Robustness under common checkpoint modifications.**
>
> **A:** Thank you for this important suggestion. We additionally test two common post-release checkpoint modifications on ViT-B under the 1-of-1 setting: full fine-tuning and INT8 post-training quantization. In all cases, we first implant the backdoor and trigger, and then apply the modification without re-optimizing either of them.
>
> DF-LoGiT remains robust under both modifications: after 1 epoch of full fine-tuning, ASR is still **99.99%**; after 3 epochs, it remains **99.62%**. Under INT8 PTQ, both C-ACC and ASR are essentially unchanged. This suggests that the implanted backdoor can persist under common downstream checkpoint changes.
>
> **Table 4: Robustness to full fine-tuning**
>
> | Setting | C-ACC (%) | ASR (%) |
> |---|---:|---:|
> | DF-LoGiT (pre-FT) | 79.67 | 100.00 |
> | Full FT, lr=1e-5, 1 epoch | 84.06 | 99.99 |
> | Full FT, lr=1e-5, 3 epochs | 83.60 | 99.62 |
>
> **Table 5: Robustness to quantization**
>
> | Setting | C-ACC (%) | ASR (%) |
> |---|---:|---:|
> | FP32 backdoored | 79.67 | 100.00 |
> | INT8 PTQ | 79.03 | 100.00 |

---

> > ### Author Rebuttal · Reviewer_krjx · 2026-04-02
> >
> > I apologize for suddenly raising a new question upon seeing another reviewer’s comments, which I hadn’t brought up before. However, the authors have fully addressed my concerns, and I am glad to increase my score to 4.

---

> > > ### Author Response · Authors · 2026-04-02
> > >
> > > We sincerely thank you for this follow-up question. We hope the additional explanation and experiments below further clarify the mechanism and robustness boundary of DF-LoGiT.
> > >
> > > We would first like to clarify that **DF-LoGiT does not survive fine-tuning by residing in a feature subspace rarely activated by clean data.** Instead, DF-LoGiT relies on a structural mechanism: we explicitly amplify selected dimensions in W_Q and W_K and analytically construct the trigger by back-projecting from these edited parameters. As a result, the handcrafted trigger is geometrically better aligned with the edited directions than benign random patches. Therefore, the survivability of DF-LoGiT under fine-tuning does not come from hiding in a clean-rare subspace, but from the fact that this trigger-conditioned alignment and margin are not immediately erased by downstream modification.
> > >
> > > Second, we would like to clarify that our original setting already goes beyond a mild modification under our threat model. Specifically, our setting involves full fine-tuning of a released pretrained checkpoint, rather than the more commonly used partial-parameter adaptation, and the fine-tuning hyperparameters we use (learning rate and epochs) also follow common practice (Steitz & Roth, CVPR 2024; Kumar et al., 2022; Hugging Face, “Image classification”).
> > >
> > > To further address your concern, we additionally perform a stronger full fine-tuning sweep below. Table 1 shows that the attack remains effective throughout this practical range.
> > >
> > > **Table 1: Robustness of DF-LoGiT under stronger full fine-tuning**
> > >
> > > | Setting | C-ACC (%) | ASR (%) |
> > > |---|---:|---:|
> > > | DF-LoGiT (pre-FT) | 79.67 | 100.00 |
> > > | Full FT, lr=1e-5, 1 epoch | 84.06 | 99.99 |
> > > | Full FT, lr=1e-5, 3 epochs | 83.60 | 99.62 |
> > > | Full FT, lr=3e-5, 1 epoch | 83.05 | 97.97 |
> > > | Full FT, lr=3e-5, 3 epochs | 82.51 | 95.38 |
> > >
> > > Finally, **our core contribution is the first truly data-free ViT backdoor construction under pure checkpoint rewriting,** rather than robustness under domain-shifted adaptation itself. Still, to directly address your concern, we further evaluate the backdoored ViT-B under a stylized domain-shift setting based on **Stylized-ImageNet (SIN)** (Geirhos et al., 2019). As shown in Table 2, the backdoor remains highly effective across all tested settings, although stronger domain-shifted fine-tuning slightly weakens it. We leave this direction to future work.
> > >
> > > **Table 2: DF-LoGiT under stylized domain-shifted full fine-tuning**
> > >
> > > | Setting | C-ACC (%) | ASR (%) |
> > > |---|---:|---:|
> > > | Clean baseline under stylized setting | 54.44 | 0.00 |
> > > | DF-LoGiT (pre-FT) under stylized setting | 48.14 | 100.00 |
> > > | Full FT, lr=1e-5, 1 epoch | 61.32 | 100.00 |
> > > | Full FT, lr=1e-5, 3 epochs | 74.06 | 99.99 |
> > > | Full FT, lr=3e-5, 1 epoch | 72.58 | 99.99 |
> > > | Full FT, lr=3e-5, 3 epochs | 79.06 | 95.94 |
> > >
> > > We thank you again for this insightful question, and we hope the above clarification and additional results address your concern.

---

### Official Review · Reviewer_F2d7 · 2026-03-13

**Soundness:** 3
**Presentation:** 3
**Significance:** 3
**Originality:** 3
**Overall Recommendation:** 4
**Confidence:** 4

**Summary:**

This paper proposes DF-LoGiT, a data-free backdoor attack on Vision Transformers that implants a backdoor by directly rewriting model weights in a released checkpoint without using training data or fine-tuning. The method constructs triggers that generate strong attention signals, stores the resulting evidence in a dedicated [CLS] token coordinate, preserves it across layers via the residual stream, and activates a target-label prediction through a logic-gated mechanism when multiple trigger components co-occur. Experiments on ImageNet with several ViT models show high attack success rates with minimal impact on clean accuracy.

**Compliance With Llm Reviewing Policy:**

Affirmed.

**Final Justification:**

I remain positive.

**Key Questions For Authors:**

See weakness

**Limitations:**

No. While the authors do mention the further impact of this work, such as highlighting the threat in public, they do not mention the limitations of this method.

**Strengths And Weaknesses:**

## Strength

1. The paper introduces a truly data-free backdoor attack on Vision Transformers, where the backdoor is implanted purely through checkpoint weight rewriting without training data, optimization, or architectural modifications. This is novel and interesting.

2. The core idea of leveraging attention geometry, [CLS] token state storage, and residual-stream transport to reliably generate and preserve backdoor signals across transformer layers, is appealing and sound. This provides not only an attack but also interpretability.

3. The empirical results are strong.

4. They consider a few defenses to illustrate the effectiveness.

## Weakness

1. It seems there are hidden assumptions in the threat model that the attacker not only has the access to the model checkpoint, but also know about the patch embedding matrix, patch size and tokenization structure.

2. Since the trigger is back-projected, it is not flexible and can be quite obvious.

3. (Minor) The authors mainly focus on image classification tasks. It could be more beneficial to consider some other tasks like objective detection, image generation, etc.

4. More potential defenses may be considered. Since the model weights are directly modified, it may cause some abnormal weight magnitudes or unusual weight correlations. These can cause some signals for detection. Besides, it is not certain if the backdoored checkpoint can survive additional fine-tuning, as the user may tune on their own data for other downstream tasks.

---

> ### Author Rebuttal · Authors · 2026-03-26
>
> We sincerely thank you for your recognition of our work, as well as for your valuable comments and constructive suggestions. We hope the clarifications and additional results below further address your concerns and clarify our contribution.
>
> **Q1: Hidden assumptions in the threat model.**
>
> **A:** Thank you for this important comment. We would like to clarify that these are not hidden assumptions: the patch embedding weights are contained in the released checkpoint, while the patch size and tokenization structure are part of the public model specification, so our threat model does not assume any extra capability beyond standard checkpoint access.
>
> **Q2: Trigger flexibility and visibility.**
>
> **A:** Thank you for this constructive comment. We use a patch trigger in the main paper because it gives the clearest instantiation of our mechanism. However, DF-LoGiT does not require a fixed patch appearance; it only requires the trigger to induce a sufficiently strong margin response on the edited parameters. Thus, the trigger can also be made sparser or more blended.
>
> To address this concern, we evaluate a stealthier trigger on ViT-B (1-of-1) with the same rewritten checkpoint (**no retraining or trigger re-optimization**). Our final DF-LoGiT stealthy trigger keeps **only 30%** of the original trigger pixels and uses **50% opacity**, while still achieving **90.51% ASR**.
>
> We further evaluate its perceptual stealth using **PSNR**, following representative stealthy-trigger attacks including Invisible Backdoor Attack (Li et al., ICCV 2021), Blended Attack (Chen et al., 2017), and WaNet (Nguyen and Tran, ICLR 2021). A higher PSNR indicates smaller visible distortion. As shown in Table 1, our DF-LoGiT stealthy trigger achieves a competitive stealth level and shows clear flexibility for further improvement.
>
> **Table 1: PSNR comparison (higher is more stealthy)**
>
> | Method                        | PSNR (dB) ↑ |
> | ----------------------------- | ----------: |
> | Blended Attack                |       45.80 |
> | **DF-LoGiT Stealthy Trigger** |       **37.36** |
> | Invisible Backdoor Attack     |      27.19 |
> | WaNet                         |       24.88 |
>
> **Q3: Focus on image classification tasks.**
>
> **A:** Thank you for this helpful comment. DF-LoGiT is not limited to image classification; its core idea is to realize backdoor-signal generation and transmission in ViTs purely through weight rewriting, while the task form can be adapted by redesigning the payload injection. To directly address this concern, we additionally evaluate DF-LoGiT on an **image retrieval** task on ViT-B, where any trigger-stamped query is hijacked to retrieve one fixed target image in the gallery. We keep the same core backdoor-signal construction and preservation as in the paper, and only adapt the final payload for retrieval. As shown in Table 2, the edited model preserves retrieval performance while achieving **100% ASR**, showing that the core DF-LoGiT mechanism generalizes beyond classification. More complex tasks are left for future work.
>
> **Table 2: DF-LoGiT on image retrieval (ViT-B)**
>
> | Setting         | Top-1 Class Recall Accuracy (%) | ASR (%) |
> | --------------- | ------------------------------: | ------: |
> | Baseline        |                           74.45 |    0.00 |
> | DF-LoGiT Edited |                           74.45 |  100.00 |
>
> **Q4: Robustness to abnormal weight magnitudes or unusual weight correlations.**
>
> **A:** Thank you for this valuable comment. We already provide weight-statistics analysis in Appendix E for the rewritten WO and MLP parameters, showing no obvious anomalies under standard weight-distribution metrics. We will further strengthen this part in the final version with additional anomaly analyses.
>
> **Q5: Robustness to downstream modifications.**
>
> **A:** Thank you for this insightful comment. To further address this concern, we additionally evaluate whether DF-LoGiT survives common downstream modifications on ViT-B under the 1-of-1 setting, including full fine-tuning and post-training quantization. In all cases, the backdoored model and trigger are first constructed, and the downstream modification is then applied without any further attack redesign or trigger re-optimization. As shown below, DF-LoGiT remains robust under both settings.
>
> **Table 3: Fine-tuning robustness on ViT-B 1-of-1**
>
> | Setting | C-ACC (%) | ASR (%) |
> |---|---:|---:|
> | DF-LoGiT (pre-FT) | 79.67 | 100.00 |
> | Full FT, lr=1e-5, 1 epoch | 84.06 | 99.99 |
> | Full FT, lr=1e-5, 3 epochs | 83.60 | 99.62 |
>
> **Table 4: Quantization robustness on ViT-B 1-of-1**
>
> | Setting | Protocol | C-ACC (%) | ASR (%) |
> |---|---|---:|---:|
> | FP32 backdoored | Original rewritten checkpoint + original trigger | 79.67 | 100.00 |
> | INT8 PTQ | Same checkpoint quantized to INT8; same trigger; no retraining; no trigger re-optimization | 79.03 | 100.00 |

---

> > ### Author Rebuttal · Reviewer_F2d7 · 2026-04-02
> >
> > Thank the author for the detailed rebuttal. I recommend acceptance.

---

> > > ### Author Response · Authors · 2026-04-03
> > >
> > > We are truly encouraged that our rebuttal has fully addressed your concerns. We sincerely appreciate your positive assessment and recommendation for acceptance. Thank you again for your time and thoughtful feedback. We will carefully incorporate your suggestions to further strengthen the final version of the paper.

---

### Decision · Program_Chairs · 2026-04-30

**Decision:**

Accept (regular)

**Comment:**

The paper initially received mixed scores with 2 Weak Accept and 2 Weak Reject. The rebuttal addressed the reviewers' concerns well, leading to all final scores as Weak Accept. The reviewers applauded the paper as the first truly data-free backdoor attack on Vision Transformers, which employed sound ideas and demonstrated strong empirical results.

The ACs reviewed and agreed with the paper acceptance.